# Membership Privacy Risks of Sharpness Aware Minimization

**Young In Kim**[1]**, Andrea Agiollo**[2]**, Pratiksha Agrawal**[1]**, Johannes O. Royset**[3]**, Rajiv Khanna**[1]
[1]Department of Computer Science, Purdue University
[2]Faculty of Electrical Engineering, Mathematics and Computer Science, TU Delft
[3]Department of Industrial and Systems Engineering, University of Southern California

## Abstract

Optimization algorithms that seek flatter minima, such as Sharpness-Aware Minimization (SAM), are credited with improved generalization and robustness to noise. We ask whether such gains impact membership privacy. Surprisingly, we find that SAM is more prone to Membership Inference Attacks (MIA) than classical SGD across multiple datasets and attack methods, despite achieving lower test error. This suggests that the geometric mechanism of SAM that improves generalization simultaneously exacerbates membership leakage. We investigate this phenomenon through extensive analysis of memorization and influence scores. Our results reveal that SAM is more capable of capturing atypical subpatterns, leading to higher memorization scores of samples. Conversely, SGD depends more heavily on majority features, exhibiting worse generalization on atypical subgroups and lower memorization. Crucially, this characteristic of SAM can be linked to lower variance in the prediction confidence of unseen samples, thereby amplifying membership signals. Finally, we model SAM under a perfectly interpolating linear regime and theoretically show that sharpness regularization inherently reduces variance, guaranteeing a higher MIA advantage for confidence and likelihood ratio attacks.

## 1 Introduction

Sharpness-Aware Minimization (SAM) has emerged as a prominent optimization technique for improving generalization in deep learning by encouraging flatter minima – i.e., similar loss values for weight perturbations of certain degree around the optima – in the loss landscape (Norton & Royset, 2021; Foret et al., 2020; Wu et al., 2020; Kim et al., 2022; Du et al., 2022; Kwon et al., 2021). Flatter optima have been linked to robustness to noise and improved test performance (Chen et al., 2023; Foret et al., 2020; Baek et al., 2024), while a tension exists between whether SAM's implicit bias is geared more towards diversity (Springer et al., 2024) or simplicity (Andriushchenko et al., 2023; Chang & Khanna, 2025) of features.

Models that generalize well are thought to rely less on memorizing specific training examples, which should also improve privacy. Consider membership inference attacks (MIAs) in which an attacker exploits the model behavior gap between training and unseen data to infer if a data point was part of the training data or not (Shokri et al., 2017). Intuitively, when a model strongly overfits (training error ≪ test error), MIA would become easier. Yeom et al. (2018) formally showed that, under certain assumptions, the advantage of a threshold-based MIA is upper bounded by the model's generalization error. In light of this, one would naturally expect that a technique like SAM – which demonstrably improves generalization – should also decrease a model's susceptibility to MIAs.

Contrary to this expectation, we find that models trained with SAM are actually *more* vulnerable to MIAs than SGD consistently across diverse datasets and attack methods, even as they achieve better generalization (see Tables 1 and 2). Furthermore, this finding challenges the notion that *flatter minima=good* from a privacy standpoint, calling for a deeper investigation into the relationship between generalization, memorization, and privacy to unearth this phenomenon both empirically and theoretically. Our work is the first to systematically demonstrate higher membership privacy leakage for a sharpness-based algorithm known to generalize better, connecting loss sharpness to MIA

| Dataset | Optimizer | NN | Confidence | Entropy | M-entropy | Test Acc |
|---------|-----------|-----|-----------|---------|-----------|----------|
| CIFAR-100 | SGD | 76.62% | 77.19% | 76.61% | 77.30% | 80.30% |
|           | SAM | 77.99% | 79.10% | 78.66% | 79.25% | 81.60% |
| CIFAR-10 | SGD | 50% | 59.37% | 59.09% | 59.51% | 96.00% |
|          | SAM | 50.08% | 61.64% | 61.64% | 61.70% | 96.72% |
| Purchase-100 | SGD | 66.00% | 66.76% | 64.78% | 67.13% | 85.50% |
|              | SAM | 66.62% | 67.30% | 65.35% | 67.54% | 85.54% |
| Texas-100 | SGD | 59.81% | 65.20% | 55.74% | 65.13% | 50.83% |
|           | SAM | 59.56% | 66.59% | 57.14% | 65.42% | 51.34% |
| EyePacs | SGD | 73.62% | 73.40% | 68.50% | 73.40% | 73.67% |
|         | SAM | 77.73% | 77.07% | 73.37% | 77.36% | 75.41% |

Table 1: Attack accuracy of direct threshold MIA on SGD and SAM showing tradeoffs in test accuracy and MIA attacks. In green we highlight the best performing model on the test set and in orange the model against which MIA is more successful. SAM is more prone to direct threshold attacks.

vulnerability. We note that there have been previous works that exhibit utility–privacy tradeoffs (Long et al., 2018; Carlini et al., 2021; Chen et al., 2022; Liu et al., 2024) or demonstrate that higher generalization does not necessarily decrease privacy leakage (Kaya & Dumitras, 2021; Del Grosso et al., 2023).

The difference in model behavior on data points that were part of the training set versus those that were not can be more precisely quantified using *memorization scores* (Feldman, 2020; Feldman & Zhang, 2020), defined via Leave-One-Out (LOO) error. Memorization scores measure the change in model performance when a specific training sample is removed, and thus serve as a proxy for how much the model has memorized that sample. Motivated by this connection, we analyze the memorization scores of samples trained with SAM and find that *SAM exhibits more memorization than SGD*, indicating a stronger reliance on individual training samples. While this increased memorization provides a plausible explanation for SAM's heightened vulnerability to MIA, it raises a key question: "How can a model that memorizes more generalize better?"

We hypothesize that the answer lies in *what* is being memorized. Under overparameterization (Allen-Zhu et al., 2019), models can learn not only noise in the data, but also atypical patterns in under-represented subpopulations through memorization—i.e., few white tiger images with numerous yellow tiger images. This distinction is important as real world datasets are known to have a long tail of such rare subclasses (Feldman, 2020). We conjecture that SAM is capable of doing more structured memorization, selectively focusing on atypical subclass patterns, which contributes positively to generalization. Corroborating this hypothesis, we observe that SAM's memorization score distribution is concentrated in the mid range, rather than the high end—which is typically associated with noise memorization (refer to Section 4.1). This suggests that SAM emphasizes samples that are neither trivially learned nor purely noisy, but instead represent rare, but generalizable sub-patterns.

To further validate this finding, we analyze influence scores—which measure the impact of individual training samples on test predictions (see Section 4.2). Our results show that, for SAM, samples corresponding to moderate memorization exert higher influence on test predictions compared to SGD, confirming that SAM's generalization gains derive from its ability to better capture rare sub-patterns. Conversely, for SGD, the lower influence of such points implies that the majority pattern is learned dominantly: since this feature is redundant across many samples, the marginal influence of any single point is diluted. These results seem to suggest more that SAM's implicit bias is towards diversity as opposed to simplicity.

We support our intuition further by introducing a novel metric that quantifies the degree of memorization involved in predicting a test sample in Section 4.3. Using this metric, we dissect SAM's performance gains and identify that SAM's improvements mostly stem from its performance on atypical test samples that depend heavily on a handful of memorized training points. Meanwhile, SGD performs slightly better on typical samples that rely more on broadly learned features. This is in

| Dataset | Attack | SGD | | | | SAM | | | |
|---|---|---|---|---|---|---|---|---|---|
| | | Test Acc | AUC | Attack Acc | TPR@.1 | Test Acc | AUC | Attack Acc | TPR@.1 |
| CIFAR-100 | RMIA | 67.7% | 90.4% | 80.8% | 21.0% | 69.1% | 91.6% | 82.2% | 23.4% |
| | LiRA | | 92.6% | 82.9% | 27.0% | | 93.7% | 84.1% | 31.0% |
| CIFAR-10 | RMIA | 92.3% | 71.4% | 63.5% | 4.8% | 93.1% | 74.9% | 65.9% | 6.7% |
| | LiRA | | 73.0% | 64.2% | 8.8% | | 76.4% | 66.7% | 12.5% |
| Purchase100 | RMIA | 76.5% | 68.8% | 62.7% | 1.5% | 77.4% | 70.2% | 63.7% | 1.7% |
| | LiRA | | 68.9% | 62.6% | 1.4% | | 70.2% | 63.4% | 1.6% |
| Texas100 | RMIA | 46.9% | 79.8% | 70.8% | 2.9% | 49.2% | 80.6% | 71.5% | 2.8% |
| | LiRA | | 80.8% | 71.3% | 6.9% | | 81.6% | 72.0% | 8.2% |

Table 2: Comparison of **online** shadow-model MIA on SGD and SAM. In green we highlight the best performing model on the test set, and in orange the model with higher privacy leakage (higher AUC, Attack Accuracy, and TPR@0.1%FPR). SAM is more prone to shadow-model attacks.

line with recent works that leverage model behavior differential between SAM and SGD for dataset pruning (Agiollo et al., 2024) or active learning (Kim et al., 2025). Stronger influence of minority samples can lead to greater membership privacy risk due to the increased retention of information an attacker can exploit.

However, what intrinsic mechanism drives this structured memorization? Results for confidence threshold attack indicates that SGD produces more predictions with extreme confidence compared to SAM. These samples reside beyond the threshold and are source of attacker's error. This suggests a distinct geometric effect: *SAM induces a shrinkage in the variance of the model's output predictions*. This property translates to structured memorization and suppression of majority sub-feature. Relying heavily on a single/few majority subclass feature to classify diverse inputs requires the model to assign large weights to that feature. Geometrically, this creates a steep decision boundary and, consequently, high output variance under perturbation. By penalizing this sharpness, SAM prohibits the amplification of the majority feature, effectively forcing the model to distribute its reliance across diverse, subclass-specific features.

Completing this conjecture, we provide a theoretical foundation for this mechanism in Section 5. We prove that the interpolating solution favored by sharpness-aware geometry inherently reduces the variance of the output logits. We then demonstrate how this variance reduction amplifies the attacker's advantage for both confidence-based and likelihood ratio attacks. Our proofs highlight a strong result: *SAM is more vulnerable at any fixed threshold*. We empirically corroborate this with ROC curves in Figure 7. Lastly, we theoretically analyze a dataset with majority and minority subclasses and show how capturing minority feature better leads to enhanced generalization (see Appendix B).

**Contributions**    In summary, our contributions are the following: *(i)* we are the first to empirically demonstrate that SAM-trained models exhibit higher membership privacy risk than SGD-trained models, serving as a cautionary tale against *flatter minima=good* notion from a privacy standpoint; *(ii)* we offer a detailed and conceptually grounded analysis of the root causes of SAM's generalization-memorization relationship, suggesting SAM's implicit bias towards diversity; *(iii)* we introduce a novel methodology to dissect generalization gains, proving that SAM's generalization gains stem from its performance on unseen atypical samples; *(iv)* we theoretically show variance shrinkage effect of interpolating sharpness-aware solutions and how it increases MIA risk for both confidence and likelihood ratio attacks; and *(v)* we theoretically formulate a data distribution composed of subclasses where stronger alignment with minority subclass features enhances generalization.

## 2    BACKGROUND & PRELIMINARIES

**Memorization & Influence scores**    For a training algorithm $\mathcal{A}$ that is used to train the model $f(\cdot)$ using dataset $\mathcal{D} = ((\mathbf{x}_1, y_1), ..., (\mathbf{x}_n, y_n))$, the amount of label memorization by $\mathcal{A}$ on a sample $(\mathbf{x}_i, y_i) \in \mathcal{D}$ is defined by Equation (1). The probability is taken over randomness of the algorithm

such as weight initialization.

$$mem(\mathcal{A}, \mathcal{D}, i) := \Pr_{f \leftarrow \mathcal{A}(\mathcal{D})} [f(\mathbf{x}_i) = y_i] - \Pr_{f \leftarrow \mathcal{A}(\mathcal{D} \backslash (\mathbf{x}_i, y_i))} [f(\mathbf{x}_i) = y_i] \tag{1}$$

Influence score of a training example $(\mathbf{x}_i, y_i)$ on test example $(\mathbf{x}'_j, y'_j)$ is defined by:

$$infl(\mathcal{A}, \mathcal{D}, i, j) = \Pr_{f \leftarrow \mathcal{A}(\mathcal{D})} [f(\mathbf{x}'_j) = y'_j] - \Pr_{f \leftarrow \mathcal{A}(\mathcal{D} \backslash (\mathbf{x}_i, y_i))} [f(\mathbf{x}'_j) = y'_j] \tag{2}$$

**Sharpness Aware Minimization (SAM)** Consider a model $f : X \rightarrow Y$ parameterized by a weight vector $\mathbf{w}$ and a per-sample loss function $l$: $W \times X \times Y \rightarrow R_+$. Given a dataset S = {$(\mathbf{x}_1, y_1)$,..., $(\mathbf{x}_n, y_n)$} sampled i.i.d. from a data distribution, the training loss is defined as $L_S(\mathbf{w}) = \sum_{i=1}^{n} l(y_i, f(\mathbf{x}_i, \mathbf{w}))/n$. Sharpness Aware Minimization combines traditional loss with sharpness term to minimize the difference between maximum loss in the vicinity (a Ball of radius $\rho$: $B(\rho)$) of the current minima. Formally, it is defined as the following:

$$L_{SAM}(w) = \min_w L_S(\mathbf{w}) + [\max_{\epsilon \in B(\rho)} L_S(\mathbf{w} + \epsilon) - L_S(\mathbf{w})] = \min_w \max_{\epsilon \in B(\rho)} L_S(\mathbf{w} + \epsilon) \tag{3}$$

### 2.1 MEMBERSHIP INFERENCE ATTACKS

Consider a victim model $f_v$ trained on dataset $D \sim \mathcal{D}$ and attack model $f_a$. In a black-box setting, an attacker infers whether a sample $(\mathbf{x}, y)$ belongs to $D$ (IN) or not (OUT). In this paper, we consider two types of attacks: *Direct threshold attacks*, which directly learn a threshold from obtained member/non-member data, such as confidence and entropy attacks; and *Shadow model attacks*, which train proxy models to calibrate membership scores, such as Likelihood Ratio Attack (LiRA) and Robust Membership Inference Attack (RMIA) (Carlini et al., 2022; Zarifzadeh et al., 2024).

We quantify privacy risk using the empirical attack accuracy, defined as the average of the true positive (TPR) and true negative rates (1-FPR):

$$\text{Acc}_{\text{MIA}} = \frac{1}{2n_{\text{m}}} \sum_{i=1}^{n_{\text{m}}} \mathbb{1}[f_{\text{a}}(\mathbf{x}_i, y_i) = 1] + \frac{1}{2n_{\text{nm}}} \sum_{j=1}^{n_{\text{nm}}} \mathbb{1}[f_{\text{a}}(\mathbf{x}_j, y_j) = 0], \tag{4}$$

where $n_m, n_{nm}$ are the counts of IN and OUT samples. Additionally, metrics such as Area Under ROC Curve (AUC) and TPR at low FPR are employed to characterize vulnerability. Further details are provided in Appendix D.2.

## 3 PRIVACY RISKS OF SAM

Inspired by the link between SAM and generalization and how MIAs should exploit poor generalization, we here scrutinize the membership privacy risk of SAM by comparing the membership attack accuracy (see Equation (4) and Table 2) of different MIAs against SAM- and SGD-trained models across five different benchmark datasets for direct threshold attacks and four different benchmark datasets for shadow model attacks.

We utilize datasets and target models that are widely employed in studies on MIAs and defenses (Yeom et al., 2018; Fang & Kim, 2024; Chen et al., 2022; Jia et al., 2019). Furthermore, we assume that the attacker has access to some portion of the training data and non-training data that it uses to train the attack models—a common assumption in the MIA literature.

**Datasets** We use CIFAR-10, CIFAR-100 and Purchase-100 along with two medical datasets Texas-100 and EyePacs. For direct threshold attacks, we follow Tang et al. (2022) to determine the partition between training and test data and to determine the subset that constitutes the attacker's prior knowledge [1]. For shadow model attacks, we use a different dataset split to account for shadow model training [2]. More details about the datasets and the experimental setup can be found in Appendix G.

---

[1] We adopt and extend the code in `https://github.com/inspire-group/MIAdefenseSELENA`
[2] We adopt and extend the code in `https://github.com/orientino/lira-pytorch`

**Methods**    For direct threshold attacks, we train a set of models and choose the one achieving highest validation accuracy. We then employ different MIA methods – namely NN-based, confidence-based, entropy-based and modified entropy-based attacks (see Appendix D.2 for a detailed formulation of each MIA) – to evaluate the attack accuracy on the target model. For shadow model attacks, we generate 256 random half-splits of the training dataset into member set and non-member set and train a model for each split. One model is chosen as the target model and all the other models are used as shadow models for reference. More details about the experimental settings can be found in Appendix H.

**Results**    *Direct Threshold Attacks* We report the attack accuracy and test accuracy for each model in Table 1. We observe that while SAM achieves higher generalization performance, it also incurs highest attack accuracy for almost all settings. To further investigate the connection between flatness of minima and membership privacy, we report the results for other sharpness-aware optimizers and custom designed optimizer that explicitly aims to find sharper minima in Section E. The results support a relationship between loss landscape geometry and membership privacy, with other optimizers exhibiting similar behavior. For confidence threshold attack, we observe that SGD model incur higher number of extremely confident predictions compared to SAM for both correctly classified and wrongly classified non-members. Because the threshold is typically set near a high value (i.e. 0.92), non-member samples with high confidence are missed by the attacker. From this observation, we conjecture that variance of the model output is an important factor in MIA risk. To confirm that these results are not model-dependent, we report an ablation study in Appendix I and verify that similar findings can be observed for different model architectures over the same datasets.

*Shadow Model Attacks* Table 2 summarizes the results for online Robust MIA (RMIA) and Likelihood Ratio Attack (LiRA), reporting Attack Accuracy, AUC, and TPR at 0.1% FPR averaged over 10 attack splits (Zarifzadeh et al., 2024; Carlini et al., 2022). Consistent with the previous results, SAM achieves higher generalization while incurring non-trivially higher privacy leakage across nearly all settings. We note that the reported test accuracies differ from Table 1 due to difference in data splits for shadow model training.

The results for offline attacks are presented in Table 4. Analogous to the online setting, SAM exhibits superior generalization paired with heightened MIA vulnerability. Since shadow model attacks represent the state-of-the-art in membership inference, these results suggest that SAM's vulnerability is not merely an artifact of global threshold shifts. Instead, it points to an intrinsic geometric property of SAM that persists even under rigorous, sample-specific SOTA attacks.

## 4    SAM LEARNS ATYPICAL SUBCLASS FEATURES MORE

To investigate the source of SAM's increased membership privacy risk, we analyze its optimization behavior through the lens of sample memorization and influence. We follow the procedure of Feldman & Zhang (2020) to compute the memorization and influence scores for SAM-trained models on CIFAR-100. These scores are computationally burdensome to compute, hence we make them publicly available[3]. We then compare these scores against the publicly available scores for SGD-trained models on the same dataset[4], enabling a direct comparison of sample-level behavior between the two optimizers.

### 4.1    SAM MEMORIZES ATYPICAL SUB-PATTERNS MORE

We first focus on comparing the memorization behavior of SAM and SGD. Figure 1(a) shows kernel density estimates of memorization scores for both models. Although the overall shapes of the distributions are similar – reflecting the long-tailed nature of the dataset –, SAM exhibits a lower density at the lowest end of the spectrum, with the mass redistributed more evenly across the rest of the range. This indicates that SAM assigns higher memorization scores more broadly, suggesting a structured memorization of more diverse patterns compared to SGD.

To further investigate this behavior, we plot the memorization scores of individual CIFAR-100 samples under both SAM and SGD in Figure 1(b). Each sample is represented as a blue dot, with its

---

[3]`https://github.com/ddioung/SAM_mem_infl_scores`
[4]`https://pluskid.github.io/influence-memorization/`

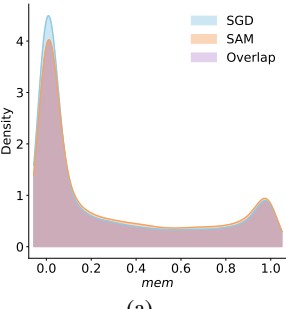 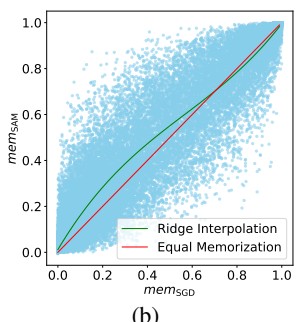 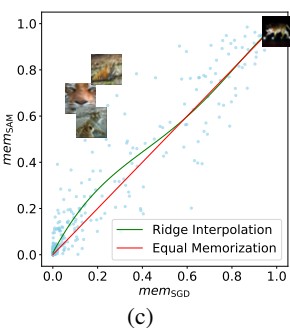

Figure 1: (a): Memorization score density plot for SAM vs SGD. SAM has less density in the lowest range, but more density spread evenly across the remaining range. (b): Memorization scores of CIFAR-100 training samples under SAM and SGD. The regression curve (in green) shows a consistent deviation from the identity line (in red), indicating that SAM memorizes a larger subset of samples in the lower score range which are likely to be atypical subclass samples. (c): Visualization of samples more memorized by SAM for the tiger class following the same setting of (b).

x- and y-coordinates corresponding to its memorization score under SGD and SAM, respectively. The red diagonal line denotes equal memorization across both optimizers. Samples above this line and to the left (top-left quadrant) are more memorized by SAM, while those below and to the right (bottom-right quadrant) are more memorized by SGD.

A regression analysis over all samples – shown via the green curve – reveals a consistent deviation from the identity – red – line, skewed towards the top-left quadrant. This indicates a systematic increase in memorization for a large subset of samples under SAM. Crucially, this deviation is not concentrated at the high end of the memorization spectrum. This finding – together with the kernel density plot – supports our hypothesis that SAM does not simply memorize pure noise samples, but rather focuses on non-dominant, atypical subclass samples that are underrepresented in the training distribution. Indeed, if SAM were picking up sample-specific noise, we would expect a sharp concentration of kernel density at the highest end of the spectrum and a deviation of the regression curve in the top-right quadrant.

Figure 1(c) illustrates this phenomenon within the *tiger* class. Samples with higher SAM memorization relative to SGD (top-left region) tend to depict clean samples containing atypical sub-patterns— e.g., close-ups of tiger heads, tigers in water, or multiple tigers in a single image. These are visually distinct yet semantically consistent with the class label. In contrast, samples with high memorization under both SAM and SGD (top-right region) often contain sample-specific noise, – e.g., a tiger with shiny paws on a pitch-black background –, which are less likely to generalize.

## 4.2 SAM INCREASES INFLUENCE OF HIGH MEMORIZATION SAMPLES

We here analyze how memorization affects the influence of training samples on test predictions, using the influence metric defined in Equation (2). Following the setup of Feldman & Zhang (2020), we first filter training–test sample pairs with influence scores above 0.2 to exclude non-influential cases. We then group training samples by memorization intervals – defined as $l < \text{mem}(\mathcal{A}, \mathcal{D}, i) < u$, with $l$ and $u$ ranging from 0 to 1 in steps of 0.05 — and, for each interval, we select the 20 training samples achieving the highest influence score on test data. This yields a distribution of influence scores of the most influential training samples conditioned on their memorization levels.

Figures 2(a) and 2(b) show the resulting distributions for SGD and SAM, respectively. As in Figure 1(b), we fit a regression curve (green line) to highlight the trends. For SGD, influence scores incur in a steep transition from lowly influent samples to highly influent points at the upper end of the memorization spectrum. This indicates SGD's reliance on a very narrow set of highly memorized samples. In contrast, SAM exhibits a smoother transition curve, with a larger set of high (and mid-to-high) memorization samples contributing more consistently to test predictions. This supports our earlier finding that SAM emphasizes a set of atypical, non-dominant subclass patterns.

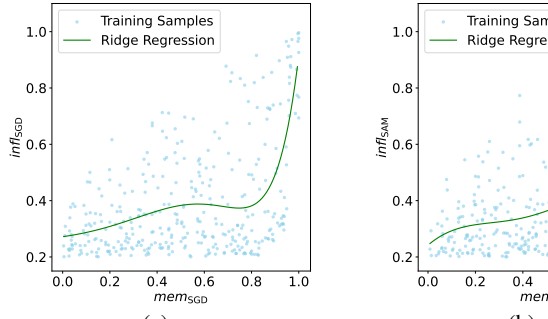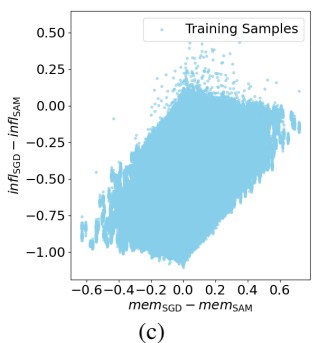

Figure 2: (a) and (b): Distribution of the influence scores of the 20 most influential training samples over each memorization interval for SGD (a) and SAM (b). The regression analysis (green lines) shows that SAM maintains a smoother influence distribution, relying more on mid-to-high (0.6 - 0.85) memorization samples (subclass features) than SGD. (c) Difference in influence scores between SAM and SGD as a function of memorization score differences. SAM downweighs influence for low-memorized samples and selectively amplifies the influence of mid-to-high memorization samples.

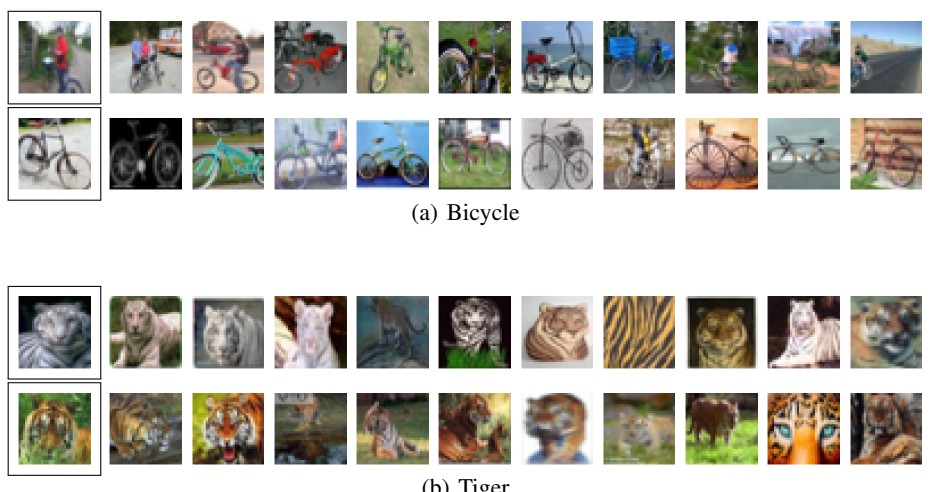

(a) Bicycle

(b) Tiger

Figure 3: Test images (boxed) from buckets 1 and 5 and their respective top-10 influential training images. For each object the top row is an image from bucket 1 and the bottom row is an image from bucket 5. For bucket 1 images (higher memorization,top row), notice that the images are atypical for their classes, and there is a near duplicate in the training data that was important for generalizing on this test image. For bucket 5 images, on the other hand, the top influential images are reminiscent of the test image at a conceptual level.

To further validate this, we examine the difference in influence scores between SAM and SGD as a function of their memorization score differences (Figure 2(c)). Training samples which are more memorized by SGD tend to have lower influence under SAM, suggesting that SAM down-weighs the influence of its low-memorized samples. Conversely, samples with similar memorization under both optimizers but higher influence under SAM tend to lie in the mid-to-high memorization range. These are precisely the samples containing atypical subpatterns that SAM selectively amplifies, confirming our intuitions.

### 4.3 SAM's Generalization Gain Comes From Higher Memorization of Sub-patterns

In this section, we dissect the generalization gains of SAM at a finer granularity by constructing a metric that divides the test data points into groups based on the amount of memorization used for predicting them. We then compare the performance on each group between SGD and SAM.

We measure the typicality of a test data point as the entropy of its corresponding training samples' influence scores. We rely on this measure since, in practice, the prediction of a typical unseen sample would be evenly influenced by numerous training data points within the same class, while atypical counterparts would be heavily influenced by a handful of training samples that are themselves also atypical. To measure the even spread, we normalize the influence scores and leverage entropy. Even influence spread would follow a more uniform distribution resulting in high entropy, while uneven spread incurs low entropy. Formally, for each test data point $i$, let $\mathcal{S}_i$ be the set of influence scores of all the training points in the same class. Let $S_{i,j}$ be the influence score of $j_{th}$ training point and $m$ be the number of training points in the same class. Then, our entropy metric $\mathcal{I}_{ent}$ is defined as:

$$\mathcal{I}_{ent}[i] = \sum_{j=1}^{m} -p_{i,j} \cdot \log p_{i,j}, \text{ where, } p_{i,j} = \frac{\mathcal{S}_{i,j}}{\sum_j \mathcal{S}_{i,j}} \tag{5}$$

We group test data points into 5 buckets in the order of lowest $\mathcal{I}_{ent}$ to highest $\mathcal{I}_{ent}$. We present some test images and their top-10 influential training images in Figure 3 from bucket 1 and bucket 5. The figure illustrates that images from bucket 1 tend to be atypical images – e.g., bicycle alongside people, white tiger, etc, – for their respective labels while images from bucket 5 tend to be more typical images—e.g., typical bicycle and yellow tiger. For quantitative verification, we plot the distribution of memorization scores of the highest influencing training points from each bucket. We observe that lower numbered buckets are associated with high memorization and vice versa (see Figures 4(b) and 4(c)). The results for other buckets interpolate between those of bucket 1 and 5, and are skipped for brevity.

We compare the generalization gains of SAM against SGD on each of these buckets and show the results in Figure 4(a)[5]. For test data points in bucket 5, SGD achieves a negligible performance gain, while for bucket 1 SAM achieves a significant gain w.r.t. SGD. Thus, the performance gains of SAM can be attributed to it correctly predicting more atypical data points which need more memorization of atypical sub-patterns to be classified correctly. For further validation, we generate a synthetic dataset and illustrate SAM's capability of learning atypical subclasses better than SGD in Appendix A. We theoretically analyze how stronger alignment with minority subclass features can lead to better generalization in Appendix B.

**Summary of experimental findings**  These results – together with those in Sections 4.1 and 4.2 – provide strong empirical evidence that *SAM's increased performance derives from better capturing atypical but informative sub-patterns, allowing predictions to be less dominated by the majority feature*. For SGD, we posit that this results in a higher output sensitivity, yielding more confident predictions on unseen data. Paradoxically, this high variance acts as a cloak for membership privacy, as it mimics the high confidence usually reserved for members. In the next section, we theoretically analyze how sharpness-regularization can inherently suppress prediction variance on unseen data.

## 5 Theoretical Analysis

In this section, we provide a theoretical foundation for the variance–shrinkage effect of geometry-aware solutions and its implication for membership inference risk. To obtain a clean and analyzable characterization of SAM's geometric bias, we study perfectly interpolating regime of overparameterized linear models. Closest to our analysis in spirit is that of Tan et al. (2022), which explains how parameter size and ridge regression affects membership risk via a variance gap. In contrast, we introduce a curvature-aligned geometry modeling SAM and show that it provably increases membership risk.

---

[5]These results do not consider image transformations (e.g. random crop, rotations), however we have also replicated the experiment with transformations, obtaining a similar trend.

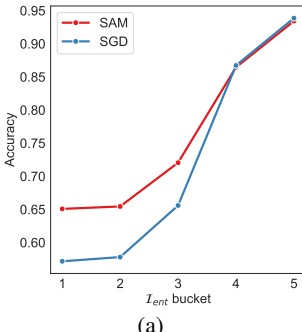 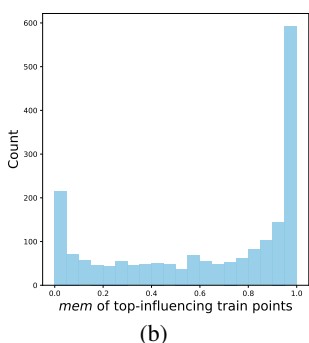 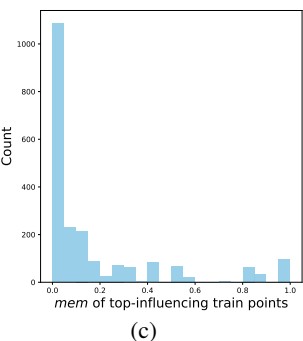

(a)  (b)  (c)

Figure 4: (a): Test accuracy on $\mathcal{I}_{ent}$ groups as evaluated by equation 5. SAM's performance gains comes from it correctly predicting more atypical data points that need memorization of atypical sub-patterns to be classified correctly. (b) and (c): Distribution of top-1 most influential training point's memorization scores for $\mathcal{I}_{ent}$ buckets 1 and 5. Testing samples falling in the lower (higher) numbered buckets are influenced by training points with higher (lower) memorization.

Our proofs are written in the regression form $X\theta = y$. We note that the connection to classification is direct. For separable binary classification with losses such as logistic or exponential, gradient descent is known to implicitly maximize the $\ell_2$ margin, and the resulting predictor is geometrically equivalent to the minimum-$\ell_2$-norm interpolator of a corresponding regression problem (Soudry et al., 2024; Gunasekar et al., 2017; Muthukumar et al., 2021). Motivated by this equivalence, we can consider classification as regression to high-magnitude targets $y_i \in \{-M, +M\}$ for a large constant $M \gg 1$ under MSE loss. In this regime, training points are interpolated to $\pm M$ and therefore have fixed high confidence, whereas test points yield outputs that fluctuate around the decision boundary (0).

**Model and notation.**   We work in finite dimension $d \gg n$. Let the population feature covariance be $\Sigma \in \mathbb{R}^{d \times d}$ with $\Sigma \succ 0$. Draw i.i.d. samples

$$x_i \sim \mathcal{N}(0, \Sigma), \qquad y_i = \theta^{*\top} x_i + \xi_i, \quad \xi_i \sim \mathcal{N}(0, \sigma_y^2), \text{ independent of } x_i, \quad i = 1, \ldots, n.$$

Let $X \in \mathbb{R}^{n \times d}$ have rows $x_i^\top$, $y = (y_1, \ldots, y_n)^\top$, and assume $\mathrm{rank}(X) = n$. Define the orthogonal projector onto the data span and the covariance matrix

$$P := X^\top (XX^\top)^{-1} X, \qquad P^2 = P = P^\top, \qquad \widehat{\Sigma} := \frac{1}{n} X^\top X, \qquad \Sigma := \mathbb{E}[xx^\top].$$

The model is defined as $f_G(x) := \widehat{\theta}_G^\top x$. In the squared-loss linear model, the population Hessian $H$ equals $\Sigma$ and the empirical Hessian $\widehat{H}$ equals $\widehat{\Sigma}$.

Now we introduce the geometries we employ to model SGD and SAM.

**Min-$G$ interpolation.**   For any symmetric positive definite matrix $G \succ 0$, consider the minimum-$G$-norm interpolant:

$$\widehat{\theta}_G := \arg\min_{\theta \in \mathbb{R}^d} \frac{1}{2} \theta^\top G \theta \quad \text{s.t.} \quad X\theta = y. \tag{6}$$

We compare the standard Euclidean case,

$$G_0 := I_d$$

with the *sharpness-aware* geometry,

$$G_\eta := I + \eta H = I + \eta \Sigma, \qquad \eta > 0,$$

Classical implicit-bias results identify standard SGD with the Euclidean case under step size and weight initialization assumptions (Zhang et al., 2017). Sharpness-aware geometry, on the other hand, reflects SAM's penalty on local sharpness, which can be shown by expanding the minimization objective. We prove that, relative to the Euclidean interpolation, the geometry $G_\eta$ strictly reduces

the variance of non-member outputs, thereby enlarging the separation between members and non-members and increasing the advantage of membership inference attacks.

We first establish that SAM strictly reduces the variance of the model's output on unseen data. By Lemma 3, for a non-member sample $X_{\text{out}}$, and the output follows $f_G(X_{\text{out}}) \sim \mathcal{N}(0, \sigma_G^2)$.

**Theorem 1** (Variance strictly decreases for SAM geometry). *Under Assumption 1, there exists $\eta_0 > 0$ such that for all $\eta \in (0, \eta_0]$,*

$$\sigma_{G_\eta}^2 < \sigma_{G_0}^2 \qquad \text{with probability } 1 - o(1).$$

*Remark* 1. Theorem 1 formalizes a simple geometric picture. The minimum-$G$ interpolant balances fitting the training span against penalizing components in directions where $G$ is large. Moving from $G_0$ to $G_\eta$ increases the penalty precisely along high-curvature directions of the Hessian. Lemma 2 shows that, under the overlap condition in Assumption 1, this reweighting strictly suppresses the $(I - P)\Sigma\widehat{\theta}_{G_0}$ component of the interpolant. Since non-member predictions depend on $\widehat{\theta}_G$ only through the quadratic form $\widehat{\theta}_G^\top \Sigma \widehat{\theta}_G$, this suppression translates directly into a strict decrease of the output variance on unseen data.

Next, we consider confidence-threshold attack and likelihood ratio attack.

**Confidence-threshold attack.** The attacker uses the confidence score $\text{Conf}_G(x) := |f_G(x)|$ and predicts "member" iff $\text{Conf}_G(x) \geq \tau$. Let $(X_{\text{in}}, Y_{\text{in}})$ be a random training pair and $X_{\text{out}} \sim \mathcal{N}(0, \Sigma)$ an independent non-member. Define attack advantage

$$\text{Adv}_G^{\text{conf}} := \sup_{\tau \geq 0} \big( \text{TPR}_G(\tau) - \text{FPR}_G(\tau) \big).$$

**Theorem 2** (SAM strictly increases confidence-based MIA advantage). *Under Assumption 1, for all sufficiently small $\eta > 0$,*

$$\text{Adv}_{G_\eta}^{\text{conf}} > \text{Adv}_{G_0}^{\text{conf}} \qquad \text{with probability } 1 - o(1).$$

*Remark* 2. For the confidence-threshold attack, interpolation implies that member confidences are geometry-invariant: the training logits are fixed (up to label noise) for any choice of $G$. Thus, changing the geometry from $G_0$ to $G_\eta$ leaves $\text{TPR}_G(\tau)$ unchanged for every threshold $\tau$, while Theorem 1 strictly reduces the non-member variance and hence lowers $\text{FPR}_G(\tau)$ at every $\tau > 0$. In other words, SAM pushes non-member scores closer to the decision boundary, sharpening the separation between the two confidence distributions.

**Likelihood Ratio (LR) attack.** The oracle LRA predicts "member" iff $\Lambda_G(f_G(x)) \geq t$. Define

$$\Lambda_G(s) := \log \frac{p_{\text{in}}(s)}{p_{\text{out}}(s; G)}, \qquad \text{Adv}_G^{\text{LR}} := \sup_{t \in \mathbb{R}} \big( \Pr(\Lambda_G(f_G(X_{\text{in}})) \geq t) - \Pr(\Lambda_G(f_G(X_{\text{out}})) \geq t) \big).$$

**Theorem 3** (SAM strictly increases LR-attack advantage). *Under Assumption 1, for all sufficiently small $\eta > 0$,*

$$\text{Adv}_{G_\eta}^{\text{LR}} > \text{Adv}_{G_0}^{\text{LR}} \qquad \text{with probability } 1 - o(1),$$

*where $G_\eta = I + \eta\Sigma$.*

*Remark* 3. The argument is similar to Theorem 2. Theorem 3 analyzes an oracle LR attacker that uses a single global IN/OUT distribution. Corollary 5 strengthens this to sample-adaptive IN/OUT distributions, matching the per-query calibration used by LiRA/RMIA. We note that while Theorem 2 and Theorem 3 state an improvement in the supremum advantage, the proof in fact shows a stronger statement: for all sufficiently small $\eta > 0$, *the entire ROC curve of the confidence-based attack under $G_\eta$ strictly dominates that under $G_0$*. This is empirically verified in Figure 7, where SAM's ROC curve is above that of SGD's for nearly the *entire* range across most settings.

## 6 CONCLUSION AND FUTURE WORK

This work investigates the mechanism behind SAM's dual effect: superior generalization coupled with heightened privacy leakage. As algorithms seeking flatter minima are widely employed, our work serves as a cautionary tale users should be aware of. Furthermore, extending our findings to devise an effective defense against Membership Inference Attacks would be beneficial to the community. Finally, future research is encouraged to further scrutinize the implicit bias of SAM.

ACKNOWLEDGMENTS

This work was supported by the European Commission under the Horizon Europe Programme as part of the projects MLSysOps (Grant Agreement #101092912) and REWIRE (Grant Agreement #101070627).

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

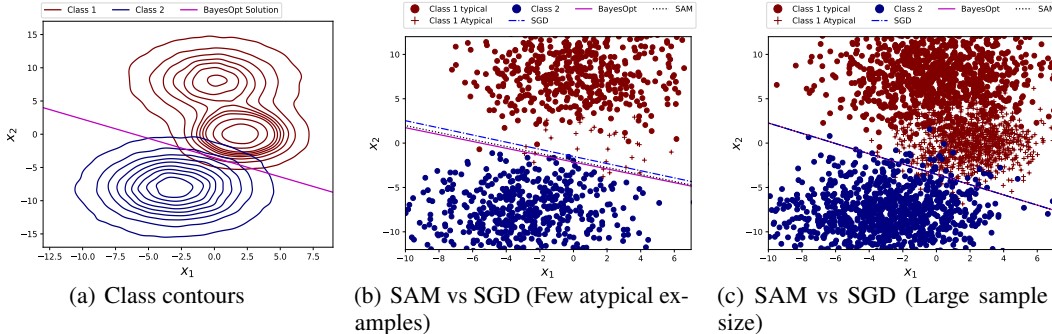

(a) Class contours   (b) SAM vs SGD (Few atypical examples)   (c) SAM vs SGD (Large sample size)

Figure 5: A synthetic construction illustrating the generalization ability of SAM over SGD for atypical examples. Fig (a) shows class density contours of a two-class, 2-dimensional classification problem, along with the bayes optimal solution. The red class has two 'clusters', one representing typical examples and one representing atypical examples. Fig (b) shows an instance of data sampled from densities shown in (a); the larger cluster of red dots represent typical examples in the red class, and the red '+' points represent a lot fewer atypical examples. SAM generalizes better than SGD in this case. Fig (c) shows that if there are enough samples generated from both typical and atypical clusters, SAM and SGD coincide with the Bayes Optimal classifier.

## A   SYNTHETIC DATASET

In this section, we provide a simple synthetic construction that illustrates how SAM can achieve better generalization performance w.r.t. vanilla SGD. The example is illustrated in Figure 5. The data is generated from two-dimensional densities illustrated in Figure 5(a). The densities are supported in two dimensions labelled as $x_1$ and $x_2$. There are two classes—the red class and the blue class. Figure 5(a) also shows the Bayes Optimal classifier. The red class has two 'clusters', one representing the typical examples (e.g. yellow tigers), and the other representing the atypical examples (e.g. white tigers). The data is sampled in such a way that we have several samples from the typical cluster, while there are only a few samples from the atypical cluster in the red class (as shown in Figure 5(b)). Figure 5(b) further shows that seeking flatter minima using the SAM optimizer learns a classifier that is closer to the bayes optimal solution than the one learnt using vanilla SGD. Therefore, SAM generalizes better. This difference in performance vanishes in Figure 5(c) when we have a large sample size for the atypical examples. This synthetic construction shows that one possible reason that SAM can perform better is its tendency to memorize atypical samples more than vanilla SGD. In other words, the gain in generalization could potentially come from those atypical data subgroups. In the next subsection, we empirically verify this conjecture for the CIFAR-100 dataset.

## B   BETTER MINORITY SUBCLASS ALIGNMENT LEADS TO HIGHER GENERALIZATION

In this section, we theoretically analyze a setting involving majority subclass samples, minority subclass samples, and pure noise samples to illustrate how an overfitting model generalizes better. Driven by the motivations in the previous sections, we discern a model by the amount of minority subclass feature it captures. Theorem 4 shows that this property leads to higher generalization. The proofs are in Section C.3.

**Definition 1** (Data Model). *Training dataset $D = \{(\mathbf{x}_i, y_i)\}_{i=1}^n$ is sampled i.i.d. from the data distribution $\mathcal{D}$. We define $\mathbf{x} = [\mathbf{x}_1, \mathbf{x}_2, \mathbf{x}_3]$, $y \in \{\pm 1\}$, $\mathbf{x}_1 \in \mathbb{R}^{d_1}$, $\mathbf{x}_2 \in \mathbb{R}^{d_2}$, $\mathbf{x}_3 \in \mathbb{R}^{d_3}$. We consider a overparameterized regime where $d_1, d_2, d_3 \gg n$. We consider a linear predictor for classification, $f_{\mathbf{w}}(\mathbf{x}) = \mathrm{sign}\{\langle \mathbf{w}, \mathbf{x} \rangle\}$ with $\mathbf{w} = [\mathbf{w}_1, \mathbf{w}_2, \mathbf{w}_3]$. We assume perfect interpolation satisfying finite margin $\forall i\ 0 \leq m_0 \leq m_i \leq M < \infty$, where margin $m_i = y_i \langle \mathbf{w}, \mathbf{x_i} \rangle$. Let $\mathcal{M}, \mathcal{S}, \mathcal{N}$ denote the majority, minority, and noise subsets, respectively. $D = \mathcal{M} \cup \mathcal{S} \cup \mathcal{N}$. We define $\sigma(z) = 1/(1 + \exp^{-z})$.*

$\mathcal{D}$ *is a mixture distribution*

$$\mathcal{D} = p_{\mathcal{M}}\,\mathcal{D}_{\mathcal{M}} + p_{\mathcal{S}}\,\mathcal{D}_{\mathcal{S}} + p_{\mathcal{N}}\,\mathcal{D}_{\mathcal{N}},$$

*with mixture weights* $p_{\mathcal{M}} + p_{\mathcal{S}} + p_{\mathcal{N}} = 1$, *and* $p_{\mathcal{M}} \gg p_{\mathcal{S}}, p_{\mathcal{N}}$. *Here,* $\mathcal{D}_{\mathcal{M}}$ *generates majority samples with* $\mathbf{x}_1 = y\boldsymbol{\mu}_1$, $\mathbf{x}_2 \sim \mathcal{N}(0, I_{d_2})$, $\mathbf{x}_3 \sim \mathcal{N}(0, I_{d_3})$ *where* $\boldsymbol{\mu}_1$ *is a fixed vector;* $\mathcal{D}_{\mathcal{S}}$ *generates minority samples with* $\mathbf{x}_1 = \boldsymbol{\nu}$, $\mathbf{x}_2 = y\boldsymbol{\mu}_2$, $\mathbf{x}_3 \sim \mathcal{N}(0, I_{d_3})$ *where* $\boldsymbol{\mu}_2$ *is a fixed vector and* $\boldsymbol{\nu}$ *is a random vector anti-aligned with* $\mathbf{w}_1$ *(*$\langle \mathbf{w}_1, \boldsymbol{\nu}\rangle < 0$*); and* $\mathcal{D}_{\mathcal{N}}$ *generates pure noise samples with* $\mathbf{x} \sim \mathcal{N}(0, I_{d_1+d_2+d_3})$. *Let* $(n_{\mathcal{M}}, n_{\mathcal{S}}, n_{\mathcal{N}})$ *denote the counts of majority, minority, and noise samples in* $S$.

Our model easily translates to a logistic regression model if we set $f_{\mathbf{w}}(\mathbf{x}) = \sigma(\langle \mathbf{w}, \mathbf{x}\rangle)$ and change the label to $y \in \{0, 1\}$. The setup in which a part of the input contains the true signal has been commonly used in previous works (Chen et al., 2023; Kou et al., 2023) but we generalize their setup by including and analyzing subclasses in the design. Concretely, one can think of a minority sample as belonging to a long-tail subgroup of the class that requires a different feature to be recognized. We formalize an anti-alignment condition: the minority features are arranged such that any model that heavily prioritizes the majority feature $u$ will perform poorly on the minority. Intuitively, fitting the minority subgroup requires the model to memorize an alternative pattern that is independent of (or even interfering with) the main decision boundary for the majority. We assume the minority subgroup is very small, so that by default a standard empirical risk minimizer might deem it negligible. This resonates with the long-tail phenomena observed in practice—i.e., a handful of unusual examples exist that a model could easily ignore without significant impact on overall training loss. However, those examples are crucial for tail generalization as they represent rare yet valid concepts that an ideal model should learn. Our assumptions reflect prior findings that real datasets contain such long-tailed subpopulations which must be memorized to achieve the best possible accuracy Feldman (2020). Formally,

**Condition 1.** *For each minority point* $i \in \mathcal{S}$, *define*

$$B_i := -y_i\langle \mathbf{w}_1, \boldsymbol{\nu}_i\rangle > 0, \qquad A := \langle \mathbf{w}_2, \boldsymbol{\mu}_2\rangle.$$

*Let* $B$ *be a random variable with CDF* $F_B(A) = \Pr(B < A)$ *such that* $B \stackrel{d}{=} B_i$ *(i.e.,* $F_B$ *is the law/distribution of the* $B_i$*'s when* $i$ *is drawn uniformly from* $\mathcal{S}$*). We assume*

$$A < B_{\max}, \qquad B_{\max} := \sup\{\, b \in \mathbb{R} \mid F_B(b) < 1 \,\}.$$

This condition assures that the majority feature still dominates globally. Furthermore, we formulate $\boldsymbol{\nu}$ as a random variable to effectively capture multiple atypical subclasses. For example, there can be a wide range of tigers that are purely white or yellowish-white. By modeling $\boldsymbol{\nu}$ as a random variable, it provides variation on the strength of anti-alignment with the majority feature.

Driven by the empirical motivations in Section 4, we define an ordering of the models as how much minority subclass alignment (MSA) they achieved. Formally,

**Definition 2** (Minority Subclass Alignment Order)**.** *Given two interpolating solutions* $\mathbf{w}^{(A)}, \mathbf{w}^{(B)}$ *trained on the same* $S$, *define*

$$A^{(A)} := \langle \mathbf{w}_2^{(A)}, y\boldsymbol{\mu}_2\rangle, \qquad A^{(B)} := \langle \mathbf{w}_2^{(B)}, y\boldsymbol{\mu}_2\rangle.$$

*We write*

$$\mathbf{w}^{(A)} \stackrel{\mathrm{MSA}}{\succcurlyeq} \mathbf{w}^{(B)}$$

*and say that* $\mathbf{w}^{(A)}$ *has higher* minority subclass alignment *than* $\mathbf{w}^{(B)}$ *if*

$$A^{(A)} \geq A^{(B)} \quad and \quad A^{(A)}, A^{(B)} < B_{\max}.$$

**Theorem 4** (Higher MSA $\implies$ Better Generalization)**.** *Let* $\mathbf{w}^{(A)}, \mathbf{w}^{(B)}$ *be two interpolating solutions trained on the same* $D$. *Under Definition 1 and regulatory conditions, if* $\mathbf{w}^{(A)} \stackrel{\mathrm{MSA}}{\succcurlyeq} \mathbf{w}^{(B)}$, *then*

$$\Pr\left(y\langle \mathbf{w}^{(A)}, \mathbf{x}\rangle > 0 \mid (\mathbf{x}, y) \sim \mathcal{D}\right) \geq \Pr\left(y\langle \mathbf{w}^{(B)}, \mathbf{x}\rangle > 0 \mid (\mathbf{x}, y) \sim \mathcal{D}\right),$$

*with strict inequality if* $F_B((A^{(B)}, A^{(A)}]) > 0$.

Theorem 4 shows that the interpolating model that aligns with the minority subclass feature more generalizes better.

## C    PROOFS

*Assumption* 1. We make the following assumptions:

(i) (Kernel overlap) Let $P := X^\top (XX^\top)^{-1} X$ be the orthogonal projector onto $\mathrm{span}(X^\top)$. The population-sharpness direction has nontrivial mass outside the data span:

$$\|(I - P)\,\Sigma\,\widehat{\theta}_{G_0}\| \ \geq \ c \ > \ 0$$

with probability $1 - o(1)$ for some constant $c$.

(ii) (Bounded interpolator) The Euclidean interpolator satisfies $\|\widehat{\theta}_{G_0}\| = O_{\mathbb{P}}(1)$.

*Remark* 4 (Justification of assumptions). Assumption 1(i) is generic in the overparameterized regime ($d \gg n$): the vector $\Sigma\widehat{\theta}_{G_0}$ is a population signal direction, while $\mathrm{span}(X^\top)$ is a random $n$-dimensional subspace. Under mild spectral regularity of $\Sigma$, their alignment is not perfect with high probability, yielding a non-negligible $(I - P)$ component.

Assumption 1(ii) is standard for benign overfitting and ensures finite test variance. If $\|\widehat{\theta}_{G_0}\|$ diverged, the Euclidean interpolator would have exploding prediction variance and thus be unstable on unseen data.

**Lemma 1.** *The unique solution of equation 6 is*

$$\widehat{\theta}_G = G^{-1} X^\top \left(XG^{-1}X^\top\right)^{-1} y, \qquad and \qquad X\widehat{\theta}_G = y.$$

*Proof.* Form the Lagrangian $L(\theta, \lambda) = \frac{1}{2}\theta^\top G\theta + \lambda^\top(X\theta - y)$. The KKT conditions are $G\theta + X^\top\lambda = 0$ and $X\theta = y$. Eliminate $\theta = -G^{-1}X^\top\lambda$ to get $-XG^{-1}X^\top\lambda = y$, hence $\lambda = -(XG^{-1}X^\top)^{-1}y$. Substituting back yields the stated $\widehat{\theta}_G$. Since $G \succ 0$, the objective is strictly convex and the solution is unique. $\qquad\square$

**Lemma 2.** *Let $\widehat{\theta}(\eta) := \widehat{\theta}_{G_\eta}$ with $G_\eta = I + \eta\Sigma$. Then*

$$\widehat{\theta}'(0) := \frac{d}{d\eta}\widehat{\theta}(\eta)\Big|_{\eta=0} = -(I - P)\,\Sigma\,\widehat{\theta}_{G_0}.$$

*Proof.* By Lemma 1,

$$\widehat{\theta}(\eta) = G_\eta^{-1}X^\top\big(XG_\eta^{-1}X^\top\big)^{-1}y.$$

Differentiate at $\eta = 0$.

(i) $\frac{d}{d\eta}G_\eta^{-1} = -G_\eta^{-1}\Sigma G_\eta^{-1}$, so $\frac{d}{d\eta}G_\eta^{-1}\big|_{\eta=0} = -\Sigma$.

(ii) Let $A(\eta) := XG_\eta^{-1}X^\top$. Then $A'(0) = -X\Sigma X^\top$ and

$$\tfrac{d}{d\eta}A(\eta)^{-1}\big|_{\eta=0} = A(0)^{-1}(X\Sigma X^\top)A(0)^{-1}.$$

Combining (i)–(ii),

$$\begin{aligned}
\widehat{\theta}'(0) &= -\Sigma\,X^\top(XX^\top)^{-1}y + X^\top(XX^\top)^{-1}(X\Sigma X^\top)(XX^\top)^{-1}y \\
&= -(I - P)\Sigma\,\widehat{\theta}_{G_0}.
\end{aligned}$$

$\square$

**Lemma 3** (Distribution of model output on non-member data). *Let $X_{\mathrm{out}} \sim \mathcal{N}(0, \Sigma)$ independently of the training set. Conditioned on $\widehat{\theta}_G$, the prediction $f_G(X_{\mathrm{out}}) = \widehat{\theta}_G^\top X_{\mathrm{out}}$ satisfies*

$$f_G(X_{\mathrm{out}}) \sim \mathcal{N}\big(0, \sigma_G^2\big), \qquad \sigma_G^2 := \widehat{\theta}_G^\top \Sigma\,\widehat{\theta}_G.$$

*Proof.* Condition on $\widehat{\theta}_G$. Since $X_{\mathrm{out}}$ is Gaussian and $f_G$ is linear in $X_{\mathrm{out}}$, the claim follows with variance $\widehat{\theta}_G^\top\Sigma\widehat{\theta}_G$. $\qquad\square$

**Lemma 4** (First derivative of non-member variance at $\eta = 0$). *With $G_\eta = I + \eta\Sigma$, let $\sigma^2(\eta) := \widehat{\theta}(\eta)^\top \Sigma \widehat{\theta}(\eta)$. Then*

$$\sigma^{2\,\prime}(0) = 2\,\widehat{\theta}_{G_0}^\top \Sigma \widehat{\theta}'(0) = -2\left\|(I-P)\Sigma\widehat{\theta}_{G_0}\right\|^2.$$

*Proof.* Differentiate: $\sigma^{2\,\prime}(0) = 2\,\widehat{\theta}_{G_0}^\top \Sigma \widehat{\theta}'(0)$ since $\Sigma$ is symmetric. Insert Lemma 2:

$$\sigma^{2\,\prime}(0) = -2\,\widehat{\theta}_{G_0}^\top \Sigma (I-P) \Sigma \widehat{\theta}_{G_0} = -2\|(I-P)\Sigma\widehat{\theta}_{G_0}\|^2,$$

using symmetry and idempotence of $(I - P)$. $\qquad\square$

**Proof of Theorem 1**

*Proof.* By Lemma 4,

$$\sigma^{2\,\prime}(0) = -2\|(I-P)\Sigma\widehat{\theta}_{G_0}\|^2.$$

Assumption 1(i) yields $\|(I-P)\Sigma\widehat{\theta}_{G_0}\| \geq c$, hence $\sigma^{2\,\prime}(0) \leq -2c^2 < 0$ with probability $1 - o(1)$.

Since $\widehat{\theta}(\eta)$ is smooth in $\eta$ for $G_\eta \succ 0$, $\sigma^2(\eta)$ is differentiable at $0$. Therefore, on the same high-probability event, there exists $\eta_0 > 0$ such that for all $\eta \in (0, \eta_0]$,

$$\sigma^2(\eta) = \sigma^2(0) + \eta\,\sigma^{2\,\prime}(0) + o(\eta) < \sigma^2(0).$$

This implies $\sigma^2_{G_\eta} < \sigma^2_{G_0}$ for all sufficiently small $\eta > 0$ with probability $1 - o(1)$. $\qquad\square$

## C.1 CONFIDENCE-THRESHOLD ATTACK

Let $\mathcal{I}_{\mathrm{in}}$ be the index set of member training points, and let $\{(x_i, y_i)\}_{i \in \mathcal{I}_{\mathrm{in}}}$ denote the training samples. For a geometry $G \succ 0$, let $\widehat{\theta}_G \in \mathbb{R}^d$ be the (interpolating) solution of equation 6. Define the signed score and confidence by

$$f_G(x) := \widehat{\theta}_G^\top x, \qquad \mathrm{Conf}_G(x) := \left|f_G(x)\right| = \left|\widehat{\theta}_G^\top x\right|.$$

We model a black-box confidence-threshold attacker as follows. Let $I$ be a random index drawn from $\mathcal{I}_{\mathrm{in}}$ (e.g., uniformly), and let the random member pair be

$$(X_{\mathrm{in}}, Y_{\mathrm{in}}) := (x_I, y_I).$$

Let $X_{\mathrm{out}} \sim \mathcal{N}(0, \Sigma)$ be an independent non-member (test) input, independent of the training set and algorithmic randomness. For a threshold $\tau \geq 0$, define

$$\mathrm{TPR}_G(\tau) := \Pr\left(\mathrm{Conf}_G(X_{\mathrm{in}}) \geq \tau\right), \tag{7}$$

$$\mathrm{FPR}_G(\tau) := \Pr\left(\mathrm{Conf}_G(X_{\mathrm{out}}) \geq \tau\right), \tag{8}$$

$$\mathrm{Adv}_G^{\mathrm{conf}} := \sup_{\tau \geq 0}\left(\mathrm{TPR}_G(\tau) - \mathrm{FPR}_G(\tau)\right). \tag{9}$$

**Lemma 5.** *For any geometry $G \succ 0$,*

$$\mathrm{Conf}_G(X_{\mathrm{in}}) = |Y_{\mathrm{in}}| \quad \text{almost surely.}$$

*Proof.* Because $\widehat{\theta}_G$ interpolates the training data, we have $\widehat{\theta}_G^\top x_i = y_i$ for every $i \in \mathcal{I}_{\mathrm{in}}$. Therefore for the random member index $I$,

$$\mathrm{Conf}_G(X_{\mathrm{in}}) = \left|\widehat{\theta}_G^\top x_I\right| = |y_I| = |Y_{\mathrm{in}}| \quad \text{a.s.}$$

$\qquad\square$

**Lemma 6.** *Conditioned on $\widehat{\theta}_G$,*

$$\mathrm{Conf}_G(X_{\mathrm{out}}) \overset{d}{=} |Z|, \;\; Z \sim \mathcal{N}(0, \sigma_G^2),$$

*Proof.* By Lemma 3 and absolute value. $\qquad\square$

**Lemma 7.** *Let $0 < \sigma_1 < \sigma_2$ and let $Z_k \sim \mathcal{N}(0, \sigma_k^2)$ for $k \in \{1, 2\}$. Then for every $\tau > 0$,*
$$\Pr(|Z_1| \geq \tau) < \Pr(|Z_2| \geq \tau).$$

*Proof.* Let $U \sim \mathcal{N}(0, 1)$. Then $Z_k \overset{d}{=} \sigma_k U$, so
$$\Pr(|Z_k| \geq \tau) = \Pr\left(|U| \geq \frac{\tau}{\sigma_k}\right).$$
The function $t \mapsto \Pr(|U| \geq t)$ is strictly decreasing on $(0, \infty)$. Since $\tau/\sigma_1 > \tau/\sigma_2$, the claim follows. $\qquad\square$

**Lemma 8.** *Assume $\Pr(|Y_{\text{in}}| > 0) > 0$ (true whenever labels have any continuous noise). Then*
$$\mathrm{Adv}_G^{\text{conf}} = \sup_{\tau > 0}\Big(\mathrm{TPR}_G(\tau) - \mathrm{FPR}_G(\tau)\Big).$$

*Proof.* At $\tau = 0$, $\mathrm{TPR}_G(0) = \mathrm{FPR}_G(0) = 1$, so the gap equals 0. Because $\Pr(|Y_{\text{in}}| > 0) > 0$, we have $\mathrm{TPR}_G(\tau) > 0$ for some $\tau > 0$. Also, by Lemma 6, $\mathrm{FPR}_G(\tau) \to 0$ as $\tau \to \infty$. Hence there exists a $\tau > 0$ such that $\mathrm{TPR}_G(\tau) - \mathrm{FPR}_G(\tau) > 0$. Therefore the supremum cannot be attained at $\tau = 0$, and we may restrict to $\tau > 0$. $\qquad\square$

**Proof of Theorem 2**

*Proof.* By Lemma 5, member confidences are geometry-invariant, so
$$\mathrm{TPR}_{G_\eta}(\tau) = \mathrm{TPR}_{G_0}(\tau) \quad \text{for all } \tau \geq 0.$$
By Theorem 1, for all sufficiently small $\eta > 0$,
$$\sigma_{G_\eta}^2 < \sigma_{G_0}^2 \quad \text{with probability } 1 - o(1).$$

Condition on this high-probability event. Lemma 6 implies $\mathrm{Conf}_{G_\eta}(X_{\text{out}}) \overset{d}{=} |Z_\eta|$ with $Z_\eta \sim \mathcal{N}(0, \sigma_{G_\eta}^2)$, and similarly for $G_0$. Then Lemma 7 yields that for every $\tau > 0$,
$$\mathrm{FPR}_{G_\eta}(\tau) = \Pr(|Z_\eta| \geq \tau) < \Pr(|Z_0| \geq \tau) = \mathrm{FPR}_{G_0}(\tau).$$
Therefore, for every $\tau > 0$,
$$\mathrm{TPR}_{G_\eta}(\tau) - \mathrm{FPR}_{G_\eta}(\tau) > \mathrm{TPR}_{G_0}(\tau) - \mathrm{FPR}_{G_0}(\tau).$$
Taking the supremum over $\tau > 0$ and using Lemma 8,
$$\mathrm{Adv}_{G_\eta}^{\text{conf}} = \sup_{\tau > 0}\Big(\mathrm{TPR}_{G_\eta}(\tau) - \mathrm{FPR}_{G_\eta}(\tau)\Big) > \sup_{\tau > 0}\Big(\mathrm{TPR}_{G_0}(\tau) - \mathrm{FPR}_{G_0}(\tau)\Big) = \mathrm{Adv}_{G_0}^{\text{conf}}.$$
This holds with probability $1 - o(1)$. $\qquad\square$

## C.2 Likelihood-ratio attack

We consider an oracle likelihood-ratio attack that uses the model output $f_G(x) = \widehat{\theta}_G^\top x$. The attacker knows the true member and non-member score distributions and performs the Neyman–Pearson likelihood-ratio test.

**Score distributions.** Recall the data model $x \sim \mathcal{N}(0, \Sigma)$ and $y = \theta^{*\top} x + \xi$ with $\xi \sim \mathcal{N}(0, \sigma_y^2)$ independent of $x$. Let $(X_{\text{in}}, Y_{\text{in}})$ be a random member training pair obtained by sampling a random index from the training set. Unconditionally over the training data draw, we have the same marginal law as a fresh sample:
$$X_{\text{in}} \sim \mathcal{N}(0, \Sigma), \qquad Y_{\text{in}} = \theta^{*\top} X_{\text{in}} + \xi.$$

**Lemma 9** (Member score is geometry-invariant Gaussian). *For any geometry $G$, the member score satisfies*
$$f_G(X_{\text{in}}) = Y_{\text{in}} \quad \text{a.s.} \qquad \text{and hence} \qquad f_G(X_{\text{in}}) \sim \mathcal{N}(0, v_{\text{in}}),$$
*where $v_{\text{in}} := \theta^{*\top} \Sigma \theta^* + \sigma_y^2$.*

*Proof.* Interpolation gives $X\widehat{\theta}_G = y$, so for every training point $\widehat{\theta}_G^\top x_i = y_i$. Thus for a random member index $I$,

$$f_G(X_{\text{in}}) = \widehat{\theta}_G^\top x_I = y_I = Y_{\text{in}} \quad \text{a.s.}$$

Unconditionally, $Y_{\text{in}} = \theta^{*\top} X_{\text{in}} + \xi$ with $X_{\text{in}} \sim \mathcal{N}(0, \Sigma)$ and $\xi \sim \mathcal{N}(0, \sigma_y^2)$ independent, so it is mean-zero Gaussian with variance $v_{\text{in}}$. $\square$

**Lemma 10** (Non-member score is geometry-dependent Gaussian). *Let $X_{\text{out}} \sim \mathcal{N}(0, \Sigma)$ be independent of the training set. Conditioned on $\widehat{\theta}_G$,*

$$f_G(X_{\text{out}}) = \widehat{\theta}_G^\top X_{\text{out}} \sim \mathcal{N}(0, v_{\text{out}}(G)), \qquad v_{\text{out}}(G) := \widehat{\theta}_G^\top \Sigma \widehat{\theta}_G.$$

*Proof.* Same argument as Lemma 3. $\square$

**Oracle likelihood-ratio test.** Let $p_{\text{in}}$ and $p_{\text{out}}$ be the densities of $\mathcal{N}(0, v_{\text{in}})$ and $\mathcal{N}(0, v_{\text{out}}(G))$, respectively. The oracle LR score is

$$\Lambda_G(s) := \log \frac{p_{\text{in}}(s)}{p_{\text{out}}(s)}.$$

**Lemma 11.** *For zero-mean Gaussians with variances $v_{\text{in}} > 0$ and $v_{\text{out}}(G) > 0$, the LR test $\Lambda_G(s) \geq t$ is equivalent to $|s| \geq \tau$ for some $\tau \geq 0$. Moreover, the optimal test for any fixed FPR is of this form.*

*Proof.* For $s \in \mathbb{R}$,

$$\Lambda_G(s) = -\frac{1}{2}\log v_{\text{in}} + \frac{1}{2}\log v_{\text{out}}(G) - \frac{s^2}{2v_{\text{in}}} + \frac{s^2}{2v_{\text{out}}(G)} = C_G + \frac{s^2}{2}\left(\frac{1}{v_{\text{out}}(G)} - \frac{1}{v_{\text{in}}}\right),$$

where $C_G$ does not depend on $s$. Thus $\Lambda_G(s) \geq t$ is equivalent to $s^2 \geq \tau^2$ for some $\tau \geq 0$, i.e. $|s| \geq \tau$. Neyman–Pearson gives optimality of the LR test. Note that as we are in a setting where members are highly confident, $v_{\text{in}} > v_{\text{out}}$. $\square$

Define the LR-attack advantage as the best achievable TPR–FPR gap over all two-sided thresholds:

$$\text{Adv}_G^{\text{LR}} := \sup_{\tau \geq 0}\Big(\Pr(|f_G(X_{\text{in}})| \geq \tau) - \Pr(|f_G(X_{\text{out}})| \geq \tau)\Big).$$

**Proof of Theorem 3**

*Proof.* By Lemma 9,
$$f_{G_\eta}(X_{\text{in}}) \sim \mathcal{N}(0, v_{\text{in}}) \quad \text{for all } \eta \geq 0,$$
so the member tail $\Pr(|f_{G_\eta}(X_{\text{in}})| \geq \tau)$ is geometry-invariant for every $\tau \geq 0$.

By Lemma 10,
$$f_{G_\eta}(X_{\text{out}}) \sim \mathcal{N}(0, v_{\text{out}}(G_\eta)), \qquad v_{\text{out}}(G_\eta) = \sigma_{G_\eta}^2.$$
Theorem 1 gives that, for all sufficiently small $\eta > 0$,

$$v_{\text{out}}(G_\eta) < v_{\text{out}}(G_0) \quad \text{with probability } 1 - o(1).$$

Condition on this high-probability event. Then Lemma 7 (applied to $|f_G(X_{\text{out}})|$) yields that for every $\tau > 0$,
$$\Pr(|f_{G_\eta}(X_{\text{out}})| \geq \tau) < \Pr(|f_{G_0}(X_{\text{out}})| \geq \tau).$$

Therefore, for every $\tau > 0$,

$$\Pr(|f_{G_\eta}(X_{\text{in}})| \geq \tau) - \Pr(|f_{G_\eta}(X_{\text{out}})| \geq \tau) > \Pr(|f_{G_0}(X_{\text{in}})| \geq \tau) - \Pr(|f_{G_0}(X_{\text{out}})| \geq \tau).$$

Taking the supremum over $\tau > 0$ on both sides gives $\text{Adv}_{G_\eta}^{\text{LR}} > \text{Adv}_{G_0}^{\text{LR}}$. (As in Lemma 8, $\tau = 0$ yields zero gap, so the supremum is attained for some $\tau > 0$.) This holds with probability $1 - o(1)$. $\square$

**Corollary 5** (Sample-adaptive LR monotonicity). *Fix a query point $z$:*

$$f_{\text{in},z} \sim \mathcal{N}(0, v_{\text{in},z}), \qquad f_{\text{out},z}(G) \sim \mathcal{N}(0, v_{\text{out},z}(G)).$$

*Let $G_a, G_b$ be two geometries such that $v_{\text{out},z}(G_a) < v_{\text{out},z}(G_b)$. Then for every false positive rate $\alpha \in (0,1)$, the optimal likelihood-ratio test at level $\alpha$ attains strictly larger true positive rate under $G_a$ than under $G_b$. Equivalently, the per-sample ROC curve under $G_a$ strictly dominates that under $G_b$, so any MIA advantage is strictly larger under $G_a$.*

*Proof.* Fix $z$ and $k \in \{a, b\}$. The LR between $\mathcal{N}(0, v_{\text{in},z})$ and $\mathcal{N}(0, v_{\text{out},z}(G_k))$ is

$$\Lambda_z(s; G_k) = \frac{1}{2} \log \frac{v_{\text{out},z}(G_k)}{v_{\text{in},z}} + \frac{s^2}{2}\Big(\frac{1}{v_{\text{out},z}(G_k)} - \frac{1}{v_{\text{in},z}}\Big).$$

If $v_{\text{in},z} \neq v_{\text{out},z}(G_k)$ then $\Lambda_z(s; G_k)$ is a strictly monotone function of $|s|$, so by the Neyman–Pearson lemma the optimal level-$\alpha$ test is equivalent to a two-sided magnitude test $|s| \geq \tau_k(\alpha)$ for some unique threshold $\tau_k(\alpha) > 0$.

Write $f_{\text{out},z}(G_k) \overset{d}{=} \sqrt{v_{\text{out},z}(G_k)}\, U$ with $U \sim \mathcal{N}(0, 1)$. The constraint $\Pr(|f_{\text{out},z}(G_k)| \geq \tau_k(\alpha)) = \alpha$ is then

$$\alpha = \Pr\Big(|U| \geq \tfrac{\tau_k(\alpha)}{\sqrt{v_{\text{out},z}(G_k)}}\Big).$$

Since $u \mapsto \Pr(|U| \geq u)$ is strictly decreasing on $(0, \infty)$ and $v_{\text{out},z}(G_a) < v_{\text{out},z}(G_b)$, this forces

$$\frac{\tau_a(\alpha)}{\sqrt{v_{\text{out},z}(G_a)}} = \frac{\tau_b(\alpha)}{\sqrt{v_{\text{out},z}(G_b)}} \quad \Rightarrow \quad \tau_a(\alpha) < \tau_b(\alpha).$$

The member distribution $f_{\text{in},z}$ is the same under $G_a$ and $G_b$, so

$$\Pr\big(|f_{\text{in},z}| \geq \tau_a(\alpha)\big) > \Pr\big(|f_{\text{in},z}| \geq \tau_b(\alpha)\big).$$

Thus at every FPR level $\alpha$ the optimal LR/LiRA test has strictly larger TPR under $G_a$, which implies strict ROC dominance and the claimed advantage comparison. $\qquad\square$

## C.3 Higher Subclass Alignment leads to higher generalization

**Lemma 12** (High-dimensional near-orthogonality). *Let $\mathbf{x}_1, \dots, \mathbf{x}_N \in \mathbb{R}^d$ have i.i.d. $\mathcal{N}(0, 1)$ entries. Then, as $d \to \infty$,*

$$\|\mathbf{x}_i\|^2 = d\,(1 + o(1)) \quad \text{and} \quad \frac{\langle \mathbf{x}_i, \mathbf{x}_j \rangle}{\|\mathbf{x}_i\|\,\|\mathbf{x}_j\|} = o(1)$$

*for each fixed $i \neq j$, with probability tending to $1$. Moreover, the two conclusions hold* uniformly *over all $i \neq j$ with probability tending to $1$ provided $\log N = o(d)$.*

*Proof.* For norms, $\|\mathbf{x}_i\|^2 \sim \chi^2(d)$. Laurent–Massart's inequality implies that for all $t > 0$,

$$\Pr\big(\|\mathbf{x}_i\|^2 - d \geq 2\sqrt{dt} + 2t\big) \leq e^{-t}, \qquad \Pr\big(d - \|\mathbf{x}_i\|^2 \geq 2\sqrt{dt}\big) \leq e^{-t}.$$

Taking $t = \varepsilon^2 d$ gives $\|\mathbf{x}_i\|^2 = d(1 \pm O(\varepsilon))$ with probability at least $1 - 2e^{-\varepsilon^2 d}$; hence $\|\mathbf{x}_i\|^2 = d(1 + o(1))$ w.h.p. and $\|\mathbf{x}_i\| = \sqrt{d} + \mathcal{O}_p(1)$.

For inner products, write $\langle \mathbf{x}_i, \mathbf{x}_j \rangle = \sum_{k=1}^d Z_k$ with $Z_k := \mathbf{x}_{i,k} \mathbf{x}_{j,k}$, which are i.i.d., mean 0, and sub-exponential. Bernstein's inequality yields

$$\Pr\big(|\langle \mathbf{x}_i, \mathbf{x}_j \rangle| \geq t\big) \leq 2 \exp\Big(-c \min\{t^2/d,\ t\}\Big)$$

for a universal $c > 0$. Taking $t = C\sqrt{d}$ shows $|\langle \mathbf{x}_i, \mathbf{x}_j \rangle| = \mathcal{O}_p(\sqrt{d})$. Combining with $\|\mathbf{x}_i\|\,\|\mathbf{x}_j\| = d(1 + o_p(1))$,

$$\frac{|\langle \mathbf{x}_i, \mathbf{x}_j \rangle|}{\|\mathbf{x}_i\|\,\|\mathbf{x}_j\|} = \mathcal{O}_p(d^{-1/2}) = o_p(1).$$

A union bound over the $\binom{N}{2}$ pairs then gives the uniform statement whenever $\log N = o(d)$, since both tails are $\exp(-\Theta(d))$. $\qquad\square$

**Noise weights**  By the representer theorem, write

$$\mathbf{w}_3 = \sum_{j=1}^{n} \beta_j \, y_j \, \mathbf{x}_{3,j}.$$

Then, for any training point $i$,

$$y_i \langle \mathbf{w}_3, \mathbf{x}_{3,i} \rangle = \beta_i \, \|\mathbf{x}_{3,i}\|^2 \; + \; \zeta_i, \qquad \zeta_i \coloneqq \sum_{j \neq i} \beta_j \, y_i y_j \, \langle \mathbf{x}_{3,j}, \mathbf{x}_{3,i} \rangle.$$

Under Lemma 12, $\|\mathbf{x}_{3,i}\|^2 = (1 \pm o(1)) \, d_3$ and the cross inner products are $o(\|\mathbf{x}_{3,i}\| \, \|\mathbf{x}_{3,j}\|) = o(d_3)$

**Condition 1.**  For each minority point $i \in \mathcal{S}$, define

$$B_i \coloneqq -y_i \langle \mathbf{w}_1, \boldsymbol{\nu}_i \rangle \; > \; 0, \qquad A \coloneqq \langle \mathbf{w}_2, \boldsymbol{\mu}_2 \rangle.$$

Let $B$ be a random variable with CDF $F_B(A) = \Pr(B < A)$ such that $B \overset{d}{=} B_i$ (i.e., $F_B$ is the law/distribution of the $B_i$'s when $i$ is drawn uniformly from $\mathcal{S}$). We assume

$$A \; < \; B_{\max}, \qquad B_{\max} \triangleq \sup\{\, b \in \mathbb{R} \mid F_B(b) < 1 \,\},$$

*Remark* 5.  For each minority sample $i$, the anti-alignment magnitude $B_i \triangleq -y_i \langle \mathbf{w}_1, \boldsymbol{\nu}_i \rangle > 0$ summarizes how strongly the majority anchor opposes the minority anchor for that sample. We assume $\{B_i\}_{i \in \mathcal{S}}$ are i.i.d. draws from a common distribution $F_B$ supported on $(0, B_{\max}]$. Unseen minority test point has margin $m' = -B + A + \zeta'$, with $B \sim F_B$, so $\Pr_{\mathcal{S}}(m' > 0) = \Pr(B < A) = F_B(A)$ (up to the $o(1)$ fluctuation $\zeta'$ from Lemma 12). Intuitively, $F_B(A)$ is the fraction of minority subclasses whose majority anti-alignment is not too strong relative to the learned shared minority signal $A$.

**Definition 3** (Generalization gap).  *The generalization gap for a training point $i$ from distribution $\mathcal{D}_{\mathcal{K}} \in \mathcal{D}_{\mathcal{S}}, \mathcal{D}_{\mathcal{M}}, \mathcal{D}_{\mathcal{N}}$ is defined as*

$$\mathrm{R}_i^{\mathcal{K}}(w) \; \coloneqq \; \Pr\{y_i \langle \mathbf{w}, \mathbf{x}_i \rangle > 0\} - \Pr\big(y' \langle \mathbf{w}, \mathbf{x}' \rangle > 0\big), \quad (\mathbf{x}', y') \sim \mathcal{D}_{\mathcal{K}}. \tag{10}$$

*Assumption* 2 (Majority alignment).  We consider models whose first block weights align with the majority subclass feature:

$$\mathbf{w}_1 \; = \; \alpha \, \boldsymbol{\mu} \qquad (\alpha > 0),$$

This captures the implicit bias of common ERM procedures (e.g., logistic regression trained by gradient descent, or minimum-$\ell_2$-norm interpolation) to align with the dominant signal in the data.

*Assumption* 3 (Majority dominance).  We assume the majority signal dominates the stochastic parts:

$$\frac{\langle \mathbf{w}_1, \boldsymbol{\mu}_1 \rangle^2}{\|\mathbf{w}_2\|^2 + \|\mathbf{w}_3\|^2} \; \to \; \infty.$$

*Assumption* 4 (Perfect interpolation and finite-margin).  A trained model $\mathbf{w}$ has finite margins. That is, there exist $0 < m_0 \le M < \infty$ such that

$$m_0 \; \le \; y_i \langle \mathbf{w}, \mathbf{x}_i \rangle \; \le \; M \qquad \forall i.$$

**Lemma 13** (Generalization gap of majority subclass samples).  *Let $i \in \mathcal{M}$. Then with probability $1 - o(1)$,*

$$\mathrm{R}_i^{\mathcal{M}}(\mathbf{w}) = o(1)$$

*Proof.*  By Assumption 4, $y_i \langle \mathbf{w}, \mathbf{x}_i \rangle > 0$, hence $\Pr\{y_i \langle \mathbf{w}, \mathbf{x}_i \rangle > 0\} = 1$.

Draw $(x', y') \sim \mathcal{D}_{\mathcal{M}}$, so $x'_1 = y' \boldsymbol{\mu}_1$, $x'_2 \sim \mathcal{N}(0, I_{d_2})$, $x'_3 \sim \mathcal{N}(0, I_{d_3})$, independent of $y'$. Then

$$y' \langle w, x' \rangle = \langle \mathbf{w}_1, \boldsymbol{\mu}_1 \rangle + y' \langle \mathbf{w}_2, x'_2 \rangle + y' \langle \mathbf{w}_3, x'_3 \rangle =: \langle \mathbf{w}_1, \boldsymbol{\mu}_1 \rangle + Z,$$

where $Z$ is a mean-zero sub-Gaussian random variable with variance proxy $\mathrm{VarProxy}(Z) = \|\mathbf{w}_2\|^2 + \|\mathbf{w}_3\|^2$, since $y' \langle \mathbf{w}_2, x'_2 \rangle \sim \mathcal{N}(0, \|\mathbf{w}_2\|^2)$ and $y' \langle \mathbf{w}_3, x'_3 \rangle \sim \mathcal{N}(0, \|\mathbf{w}_3\|^2)$ are independent and centered.

For any $a > 0$, a standard sub-Gaussian tail bound yields

$$\Pr\{Z \leq -a\} \leq \exp\Big(-\frac{a^2}{2(\|\mathbf{w}_2\|^2 + \|\mathbf{w}_3\|^2)}\Big).$$

Taking $a = \langle \mathbf{w}_1, \boldsymbol{\mu}_1 \rangle$,

$$\Pr\{y'\langle w, \mathbf{x}'\rangle \leq 0\} = \Pr\{Z \leq -\langle \mathbf{w}_1, \boldsymbol{\mu}_1 \rangle\} \leq \exp\Big(-\frac{\langle \mathbf{w}_1, \boldsymbol{\mu}_1 \rangle^2}{2(\|\mathbf{w}_2\|^2 + \|\mathbf{w}_3\|^2)}\Big).$$

Hence

$$\Pr\{y'\langle w, \mathbf{x}'\rangle > 0\} \geq 1 - \exp\Big(-\frac{\langle \mathbf{w}_1, \boldsymbol{\mu}_1 \rangle^2}{2(\|\mathbf{w}_2\|^2 + \|\mathbf{w}_3\|^2)}\Big).$$

By Assumption 3,

$$\mathrm{R}_i^{\mathcal{M}}(\mathbf{w}) = 1 - \Pr\{y'\langle \mathbf{w}, \mathbf{x}'\rangle > 0\} \leq \exp\Big(-\frac{\langle \mathbf{w}_1, \boldsymbol{\mu}_1 \rangle^2}{2(\|\mathbf{w}_2\|^2 + \|\mathbf{w}_3\|^2)}\Big) = o(1)$$

$\square$

**Lemma 14** (Generalization gap of minority subclass samples). *Let $i \in \mathcal{S}$.*

$$\mathrm{R}_i^{\mathcal{S}}(\mathbf{x}) = 1 - F_B(A) \quad \text{(up to $o(1)$ terms)}.$$

*Proof.* By Assumption 4, $y_i\langle \mathbf{w}, \mathbf{x}_i \rangle > 0$, hence $\Pr\{y_i\langle \mathbf{w}, \mathbf{x}_i \rangle > 0\} = 1$.

Draw $(\mathbf{x}', y') \sim \mathcal{D}_{\mathcal{S}}$. $\mathbf{x}_3'$ is independent of $\{\mathbf{x}_{3,j}\}$ and mean-zero; hence $y\langle \mathbf{w}_3, \mathbf{x}_3' \rangle = \sum_j \beta_j y y_j \langle \mathbf{x}_{3,j}, \mathbf{x}_3' \rangle$ is a mean-zero fluctuation with variance vanishing relative to $\|\mathbf{x}_3'\|^2$ ; set this fluctuation to $\zeta' = o_{\mathbb{P}}(1)$ using Lemma 12. Then, $y'\langle \mathbf{w}, \mathbf{x}' \rangle = -B + A + \zeta'$.

$$\Pr\big(y'\langle w, \mathbf{x}'\rangle > 0\big) = \Pr(B < A - \zeta') \in \big(F_B(A - \varepsilon), F_B(A + \varepsilon)\big).$$

Letting $\varepsilon \downarrow 0$ gives the stated identities up to $o(1)$. $\square$

**Lemma 15** (Generalization gap for pure noise samples). *Let $i \in \mathcal{N}$ (pure noise).*

$$\mathrm{R}_i^{\mathcal{N}}(\mathbf{w}) = \frac{1}{2}$$

*Proof.* By Assumption 4, $y_i\langle \mathbf{w}, \mathbf{x}_i \rangle > 0$, hence $\Pr\{y_i\langle \mathbf{w}, \mathbf{x}_i \rangle > 0\} = 1$.

For an unseen noise $(\mathbf{x}', y') \sim \mathcal{D}_{\mathcal{N}}$, each term $y'\langle \mathbf{w}_k, \mathbf{w}_k' \rangle$ ($k = 1, 2, 3$) is a centered continuous symmetric random variable (linear form of a mean-zero isotropic vector, independent of $y'$). The sum remains centered and symmetric; hence $\Pr(y'\langle \mathbf{w}, \mathbf{x}' \rangle > 0) = 1/2$. $\square$

### Proof of Theorem 4

*Proof.* By Lemma 14, minority test accuracy equals $F_B(A)$. This is monotone increasing in $A$. The majority subclass and noise samples yield the same result for both models by Lemmas 13 and 15. $\square$

## D   ADDITIONAL RELATED WORKS

### D.1   CONNECTION OF FLATTER MINIMA WITH GENERALIZATION GAP

There have been numerous studies (Foret et al., 2020; Izmailov et al., 2018; Cha et al., 2021; Norton & Royset, 2021; Wu et al., 2020) which account for the worst-case empirical risks within neighborhoods in parameter space. Diametrical Risk Minimization (DRM) was first proposed by (Norton & Royset, 2021) and they asserted that the practical and theoretical performance of Empirical Risk Minimization (ERM) tends to suffer when dealing with loss functions that exhibit poor behavior characterized by large Lipschitz moduli and spurious sharp minimizers. They tackled this concern by employing DRM, which offers generalization bounds that are unaffected by Lipschitz moduli, applicable to

both convex and non-convex problems. Another algorithm that improves generalization is Sharpness Aware Minimization (SAM) (Foret et al., 2020) which performs gradient descent while regularizing for the highest loss in the neighborhood of radius $\rho$ of the parameter space. (Izmailov et al., 2018) proposed Stochastic Weight Averaging (SWA) that performs averaging of weights with a cyclical or constant learning rate which leads to better generalization than conventional training. They also prove that the optima chosen by the single model is in fact a flatter minima than the SGD solution. Further, (Cha et al., 2021) argues that simply performing the Empirical Risk Minimization (ERM) is not enough to achieve at a good generalization, in particular, domain generalization. Hence, they introduce SWAD which seeks for flatter optima and hence, will generalize well across domain shifts.

### D.2 DIFFERENT MEMBERSHIP INFERENCE ATTACKS

There are many variants of Direct Single-query attacks (DSQ) based on the approach of the attack and below we describe the ones used in our experiments:

**NN-based attack (Shokri et al., 2017; Tang et al., 2022; Nasr et al., 2018)** This is the first MIA proposed by Shokri et al. (2017) where they use a binary classifier to distinguish between the training members and the non-members using the victim model's behavior on these data points. The adversary can utilize the prediction vectors from the target model and incorporate them along with the one-hot encoded ground truth labels as inputs. Then, they can construct a neural network ($I_{NN}$) called attack model.

**Confidence-based attack (Yeom et al., 2020; Salem et al., 2018; Song & Mittal, 2021)** If the highest prediction confidence of an input record exceeds a predetermined threshold, the adversary considers it a member; otherwise, it is inferred as a non-member. This approach is based on the understanding that the target model is trained to minimize prediction loss using its training data, implying that the maximum confidence score of a prediction vector for a training member should be near 1. The attack $I_{conf}$ is defined as follows:

$$I_{conf}\hat{p}(y|\mathbf{x}) = \mathbb{1}(\max \hat{p}(y|\mathbf{x}) \geq \tau) \tag{11}$$

Here, $\mathbb{1}(\cdot)$ is an indicator function which returns 1 if the predicate inside it holds True else the function evaluates to 0.

**Entropy-based attack (Nasr et al., 2019; Song & Mittal, 2021; Tang et al., 2022)** When the prediction entropy of an input record falls below a predetermined threshold, the adversary considers it a member. Conversely, if the prediction entropy exceeds the threshold, the adversary infers that the record is a non-member. This inference is based on the observation that there are notable disparities in the prediction entropy distributions between training and test data. Typically, the target model exhibits higher prediction entropy on its test data compared to its training data. The entropy of a prediction vector $p(\hat{y}|x)$ is defined as follows:

$$H(p(\hat{y}|\mathbf{x})) = -\sum_i (p_i log(p_i)) \tag{12}$$

where $p_i$ is the confidence score in $p(\hat{y}|\mathbf{x})$. Then, the attack $I_{entr}$ is given as:

$$I_{entr}(\hat{p}(y|\mathbf{x}), y) = \mathbb{1}(H(p(\hat{y}|x)) \leq \tau) \tag{13}$$

**Modified entropy-based attack (Song & Mittal, 2021)** Song et al.[15] introduced an enhanced prediction entropy metric that integrates both the entropy metric and the ground truth labels. The modified entropy metric tends to yield lower values for training samples compared to testing samples. To infer membership, either a class-dependent threshold $\tau_y$ or a class-independent threshold $\tau_{attack}$ is applied.

$$I_{Mentr}(\hat{p}(y|\mathbf{x}), y) = \mathbb{1}(Mentr(p(\hat{y}|\mathbf{x})) \leq \tau_y) \tag{14}$$

where $Mentr(p(\hat{y}|\mathbf{x}))$ for (x,y) data sample is given by combination of entropy information and ground truth label as:

$$Mentr(p(\hat{y}|\mathbf{x})) = -((1 - p(\hat{y}|\mathbf{x})_y)log(p(\hat{y}|\mathbf{x})_y) - \sum_{i \neq y}(p(\hat{y}|\mathbf{x})_i log(1 - p(\hat{y}|\mathbf{x})_i))) \tag{15}$$

**Likelihood Ratio Attack (LiRA) (Carlini et al., 2022)** LiRA is a shadow-model based single–query attack that explicitly models the distributions of a scalar score for members and non-members and then performs a likelihood ratio test. For a sample $(\mathbf{x}, y)$, the attacker first defines a one-dimensional score $s(\mathbf{x}, y)$ from the target model, typically the negative cross-entropy loss or the (log-)confidence on the true label $y$. Using multiple shadow models trained with and without $(\mathbf{x}, y)$, the attacker estimates two score distributions: one for members (IN) and one for non-members (OUT). In practice, LiRA fits parametric Gaussians

$$s(\mathbf{x}, y) \mid \text{IN} \sim \mathcal{N}(\mu_{\text{in}}, \sigma_{\text{in}}^2), \qquad s(\mathbf{x}, y) \mid \text{OUT} \sim \mathcal{N}(\mu_{\text{out}}, \sigma_{\text{out}}^2),$$

and computes the log-likelihood ratio

$$\Lambda_{\text{LiRA}}(\mathbf{x}, y) = \log \frac{\phi\big(s(\mathbf{x}, y); \mu_{\text{in}}, \sigma_{\text{in}}^2\big)}{\phi\big(s(\mathbf{x}, y); \mu_{\text{out}}, \sigma_{\text{out}}^2\big)},$$

where $\phi(\cdot; \mu, \sigma^2)$ denotes the Gaussian density. The LiRA decision rule is then

$$I_{\text{LiRA}}(\hat{p}(y|\mathbf{x}), y) = \mathbb{1}\big(\Lambda_{\text{LiRA}}(\mathbf{x}, y) \geq \tau_{\text{LiRA}}\big), \tag{16}$$

for some threshold $\tau_{\text{LiRA}}$ chosen to trade off between TPR and FPR. In the "online" variant, the attacker fits both IN and OUT distributions from shadow models; in the "offline" variant, only the OUT distribution is estimated and low likelihood under the OUT model is treated as evidence of membership.

**Robust MIA (RMIA) (Zarifzadeh et al., 2024)** RMIA reframes membership inference as a calibrated hypothesis test based on a *pairwise likelihood ratio* between a query $(\mathbf{x}, y)$ and many population samples $(\mathbf{z}, y_{\mathbf{z}})$. For each pair $(\mathbf{x}, \mathbf{z})$, RMIA compares how the (approximate) probability of $\mathbf{x}$ and $\mathbf{z}$ change when conditioning on the event that $\mathbf{x}$ was used to train the target model. Concretely, RMIA defines

$$\text{LR}(\mathbf{x}, \mathbf{z}) \;=\; \frac{\Pr(\mathbf{x} \mid \theta)}{\Pr(\mathbf{z} \mid \theta)} \bigg/ \frac{\Pr(\mathbf{x})}{\Pr(\mathbf{z})},$$

where $\Pr(\cdot \mid \theta)$ denotes the target model's likelihood and $\Pr(\cdot)$ is a population prior. Intuitively, if including $\mathbf{x}$ in the target training set fits $\mathbf{x}$ disproportionately better than many other population points $\mathbf{z}$, the ratio $\text{LR}(\mathbf{x}, \mathbf{z})$ becomes large. RMIA samples many $\mathbf{z}$ from the population and defines a robust membership score

$$R(\mathbf{x}) \;=\; \frac{1}{|n_z|} \sum_{\mathbf{z}} \mathbb{1}\big(\text{LR}(\mathbf{x}, \mathbf{z}) > \gamma\big),$$

where $\gamma$ is a fixed pairwise LR threshold and $n_z$ is number of population (non-member) samples. The attack then declares membership if $R(\mathbf{x})$ exceeds a global threshold $\tau$; by sweeping $\tau$ one obtains a calibrated ROC curve, and for a chosen FPR one can directly pick the corresponding $\tau$. In the *offline* mode, all reference models are OUT models trained once on population data; in the *online* mode, the attacker additionally trains IN reference models that explicitly include $\mathbf{x}$ in their training set, which yields a more accurate approximation of the conditional likelihoods but is more computationally expensive.

## E    OTHER SHARPNESS-AWARE OPTIMIZERS

In this section, we discuss other variants of SAM, namely Adaptive SAM (ASAM) (Kwon et al., 2021), Guided SAM (GSAM) (Zhuang et al., 2022), and custom designed optimizer, namely Sharp. Sharp objective is designed to explicitly find a sharper minima. The objective function of Sharp is,

$$\mathcal{L}_{\text{Sharp}}(\mathbf{w}) = L(\mathbf{w}) - \beta \max_{\epsilon \in B(\rho)} L(\mathbf{w} + \epsilon). \tag{17}$$

This objective can be seen as minimizing the loss at current $w$ while maximizing the loss in the vicinity. We empirically verify that this objective does lead to a sharper minima measuring its hessian trace. Results and discussion about Sharp are available in Appendix H.4.

The results on CIFAR10, CIFAR100, Purchase100, and Texas100 are reported in Table 3. Other sharpness-aware optimizers are shown to achieve similar generalization gain, albeit at the cost of higher membership attack accuracy. On the other hand, optimizer that explicitly looks for a sharp minima does worse in terms of generalization, but has better membership privacy.

| Dataset | Optimizer | NN | Confidence | Entropy | M-entropy | Test Acc |
|---------|-----------|-----|-----------|---------|-----------|----------|
| CIFAR-100 | SGD | 76.62% | 77.19% | 76.61% | 77.30% | 80.30% |
| | Sharp | 57.62% | 59.69% | 57.88% | 59.69% | 76.14% |
| | ASAM | 78.92% | 79.22% | 78.86% | 79.31% | 81.80% |
| | GSAM | 78.63% | 79.23% | 79.00% | 79.23% | 82.16% |
| CIFAR-10 | SGD | 50.00% | 59.37% | 59.09% | 59.51% | 96.00% |
| | Sharp | 50.22% | 52.86% | 52.47% | 52.78% | 92.86% |
| | ASAM | 50.48% | 61.39% | 61.20% | 61.32% | 96.66% |
| | GSAM | 50.00% | 61.46% | 61.38% | 61.54% | 96.64% |
| Purchase100 | SGD | 66.00% | 66.76% | 64.78% | 67.13% | 85.50% |
| | Sharp | 59.58% | 60.96% | 58.04% | 61.16% | 84.31% |
| | ASAM | 66.85% | 66.84% | 65.39% | 67.03% | 85.54% |
| | GSAM | 67.45% | 67.72% | 66.51% | 67.87% | 85.82% |
| Texas100 | SGD | 59.81% | 65.20% | 55.74% | 65.13% | 50.83% |
| | Sharp | 51.11% | 59.89% | 53.46% | 59.36% | 49.97% |
| | ASAM | 60.92% | 67.50% | 58.80% | 67.10% | 53.17% |
| | GSAM | 54.89% | 67.07% | 57.93% | 67.13% | 52.04% |

Table 3: Attack accuracy of direct threshold MIA on SGD, Sharp, ASAM, and GSAM. In green we highlight the best performing model on the test set and in orange the model against which MIA is more successful. Analogous to SAM, optimization methods that improve generalization (ASAM, GSAM) through finding flatter minima tend to be more prone to direct threshold attacks, while optimization that looks for sharp minima instead is more robust to MIA attack while being worse off in generalization.

| Dataset | Attack | SGD | | | | SAM | | | |
|---------|--------|---------|-----|-----------|--------|---------|-----|-----------|--------|
| | | Test Acc | AUC | Attack Acc | TPR@.1 | Test Acc | AUC | Attack Acc | TPR@.1 |
| CIFAR-100 | RMIA | 67.7% | 86.8% | 77.3% | 17.3% | 69.1% | 87.7% | 78.1% | 18.9% |
| | LiRA | | 76.2% | 71.9% | 16.9% | | 77.8% | 73.2% | 19.4% |
| CIFAR-10 | RMIA | 92.3% | 69.4% | 62.3% | 4.3% | 93.1% | 72.7% | 64.6% | 5.7% |
| | LiRA | | 54.1% | 55.9% | 4.1% | | 58.7% | 58.3% | 7.0% |
| Purchase100 | RMIA | 76.5% | 68.7% | 62.6% | 1.7% | 77.4% | 70.2% | 63.6% | 1.9% |
| | LiRA | | 52.9% | 53.4% | 0.1% | | 53.7% | 54.2% | 0.1% |
| Texas100 | RMIA | 46.9% | 74.8% | 67.4% | 3.6% | 49.2% | 78.8% | 69.8% | 5.6% |
| | LiRA | | 56.9% | 58.3% | 0.8% | | 61.7% | 61.5% | 2.6% |

Table 4: Comparison of **offline** shadow model MIA on SGD and SAM. In green we highlight the best performing model on the test set, and in orange the model with higher privacy leakage (higher AUC, Attack Accuracy, and TPR@0.1%FPR).

## F    OFFLINE SHADOW MODEL ATTACKS

We report the results for offline shadow model attacks in Table 4. The results are in line and support the finding that SAM tends to incur higher membership privacy leakage. Excluding cases for tabular datasets where TPR at 0.1% FPR is near zero for both models, SAM has higher values for all other attack metrics. Consistent with the literature, RMIA is more effective for offline setting compared to LiRA (Zarifzadeh et al., 2024). For online setting, we use a different experimental setup (WideResNet targets and shadows, our own training pipeline, and a different choice of auxiliary z-points). In this setting, LiRA is slightly stronger than RMIA across most metrics (Table 3), but the gaps are modest and much smaller than those reported for the offline comparison in Zarifzadeh et al. (2024). We therefore view our results as broadly consistent with prior work: RMIA and LiRA are competitive state-of-the-art shadow-model attacks, and their exact ranking can depend on architectural and training choices. Our main conclusions – in particular, that SAM consistently increases vulnerability to both LiRA and RMIA compared to SGD – are unaffected by these small differences.

# G  DATASETS

Here we introduce the four benchmark datasets used in the experiments and they have been widely used in prior works on MI attacks:

**CIFAR-10**   [6] This is a benchmark dataset for image classification task. The dataset consists of 60,000 color images of 32x32 size. There are 6,000 images from 10 classes where 5,000 images per class belong to the training dataset and 1,000 images per class belong to the test dataset.

**CIFAR-100**   [7] The dataset is designed to be more challenging than CIFAR-10 as it contains a greater number of classes and more fine-grained distinctions between objects. There are a total of 60,000 images from 100 classes. Each subclass consists of 600 images, and within each subclass, there are 500 training images and 100 testing images. This distribution ensures a balanced representation of each class in both the training and testing sets.

**Purchase-100**   [8] This a 100 class classification task with 197,324 data samples and consists of 600 binary feature; each dimension corresponds to a product and its value states if corresponding customer purchased the product; the corresponding label represents the shopping habit of the customer. We use the pre-processed and simplified version provided by Shokri et al. (2017) and used by Tang et al. (2022).

**Texas-100**   [9] This dataset is based on the Hospital Discharge Data public files with information about inpatients stays in several health facilities released by the Texas Department of State Health Services from 2006 to 2009. We used a prepossessed and simplified version of this dataset provided by (Shokri et al., 2017) and used by (Tang et al., 2022) which is composed of 67,330 data samples with 6,170 binary features. Each feature represents a patient's medical attribute like the external causes of injury, the diagnosis and other generic information.The classification task is to classify patients into 100 output classes which represent the main procedure that was performed on the patient.

**EyePacs**   [10] The pre-processed version of this dataset is obtained from Kaggle and it was originally used for a Diabetic Retinopathy Detection challenge. The dataset consists of 88,702 colour fundus images, including 35,126 samples for training and 53,576 samples for testing. The images were captured under various conditions by various devices at multiple primary care sites throughout California and elsewhere. For each subject, two images of the left and right eyes were collected, with the same resolution. A clinician was asked to rate each image for the presence of DR with a scale of 0–4 according to the Early Treatment Diabetic Retinopathy Study (ETDRS) scale. Note that for this dataset only training set (35k images) is used since the labels for testing set is not publicly available. The images in the dataset vary in their image resolution and we resized all the images to 128x128 pixels for our experiments.

# H  EXPERIMENTAL SETUP

## H.1  MODELS

For CIFAR-100 and CIFAR-10, we use WideResNet (WRN) (Zagoruyko & Komodakis, 2016) with 16 layer depth and 8 as width factor. For Purchase-100 and Texas-100, we follow the setting in  Tang et al. (2022) and use a 4-layer fully connected neural network with layer sizes [1024, 512, 256, 100]. For EyePacs, we use ResNet-18.

---

[6] https://www.cs.toronto.edu/ kriz/cifar.html

[7] https://www.cs.toronto.edu/ kriz/cifar.html

[8] https://www.kaggle.com/c/acquire-valued-shoppers-challenge

[9] https://www.dshs.texas.gov/THCIC/Hospitals/Download.shtm.

[10] https://www.kaggle.com/datasets/mariaherrerot/eyepacspreprocess

## H.2 $\mathcal{I}_{ent}$ EXPERIMENT

We discuss how test data points were grouped into 5 buckets according to different $\mathcal{I}_{ent}$ levels. Bucket 5 contains highest $\mathcal{I}_{ent}$ level, and is composed of test points where all 500 training points have 0 influence score. This means that the prediction output for that test point does not change had the model been trained without any one particular training data point. Because influence scores for all training points are equal, these test points have highest $\mathcal{I}_{ent}$ [11]. Figure 6 displays distribution of $\mathcal{I}_{ent}$ for remaining test data points. We group those above 6.1 into bucket 4. For the rest of the points, we calculate the mean and standard deviation and use them for grouping. We group points below $-0.4\sigma$ from the mean into bucket 1, points between $-0.4\sigma$ and $0.4\sigma$ into bucket 2, and points above $0.4\sigma$ into bucket 3. Final number of test points in each buckets are [Bucket 1: 1924, Bucket 2: 2996, Bucket 3: 2392, Bucket 4: 535, Bucket 5: 2153]. For SAM's buckets, final number of test points are [Bucket 1: 1913, Bucket 2: 3181, Bucket 3: 2548, Bucket 4: 502, Bucket 5: 1856]. Number of overlapping indices were [Bucket 1: 1116, Bucket 2: 1625, Bucket 3: 1199, Bucket 4: 133, Bucket 5: 1678].

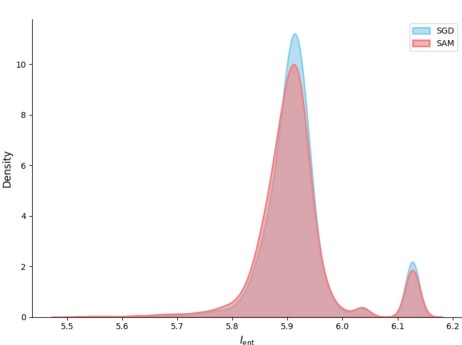

Figure 6: $\mathcal{I}_{ent}$ distribution excluding bucket 5 for SGD and SAM

## H.3 ATTACK SETUP & SIZE OF DATA SPLITS

We adopt the attack setting from (Tang et al., 2022; Nasr et al., 2018) to determine the partition between training data and test data and to determine the subset of the training and test data that constitutes attacker's prior knowledge for CIFAR-100, Purchase-100 and Texas-100 datasets. We use similar strategy to determine the data split for CIFAR-10. Specifically, the attacker's knowledge corresponds to half of the training and test data, and the MIA success is evaluated over the remaining half. For shadow model attacks, the total sample pool used is 50000 for CIFAR10 and CIFAR100, 40000 for Purchase100, and 20000 for Texas100. For RMIA attack, we used $\gamma = 1$ and selected all of the $z$ samples within the training pool that were not part of the target model's training set. On CIFAR10, for example, number of $z$ samples was 25000.

## H.4 HYPERPARAMETER TUNING AND EMPIRICAL VALIDATION OF FLATNESS FOR SHARP

For the Sharp objective (see Equation (17)), we fine-tuned $\beta$ and $\rho$ which result in a model that exhibits sufficient difference in test accuracy and sharpness of the minima compared to SAM and SGD. The final hyperparameters of the model reported were $\rho = 0.01, \beta = 0.6818$ for CIFAR-100 and CIFAR-10, $\rho = 0.01, \beta = 0.83$ for Purchase-100, $\rho = 0.001, \beta = 0.513$ for Texas-100, and $\rho = 0.001, \beta = 0.18$ for EyePacs. To verify that Sharp actually finds a sharper minima, we computed the trace of the hessian matrix using Hutchinson's method for SGD, SAM, and Sharp models on CIFAR-100. The results are in Table 5. Higher trace indicates a sharper minima and vice versa. The trace is the largest for Sharp and smallest for SAM.

| Optimizer | Trace of the Hessian |
| --- | --- |
| Sharp | 1556.54 |
| SGD | 307.87 |
| SAM | 84.18 |

Table 5: Comparison of Hessian trace values across methods.

### H.4.1 BALL OF RADIUS $\rho$

For SAM loss, sharp minima loss, and our proposed loss, we approximate the maximum loss in the ball of radius $\rho$ around the minima. Norton & Royset (2021)

---

[11]When actually calculating $\mathcal{I}_{ent}$ with Equation (5), this evaluates to 0 due to probability normalization, but represents highest value.

| Dataset | Model | Optimizer | Test Acc | Best Attack Acc |
|---------|-------|-----------|----------|-----------------|
| CIFAR-100 | Resnet18 | SGD | 78.42% | 74.31% |
| | | SAM | 78.74% | 77.45% |
| | InceptionV4 | SGD | 77.44% | 77.22% |
| | | SAM | 79.60% | 80.82% |
| CIFAR-10 | Resnet18 | SGD | 95.18% | 57.90% |
| | | SAM | 96.16% | 60.05% |
| | InceptionV4 | SGD | 94.26% | 61.60% |
| | | SAM | 95.76% | 64.41% |

Table 6: Privacy vs generalization tradeoff for SAM and SGD using the InceptionV4 and ResNet-18 models. In green we highlight the best performing model on the test set and in orange the model against which MIA is more successful. The results confirm the architecture-independent nature of our findings.

found that the type of norm that is used for defining the ball has large impact along with actual $\rho$ value. For all our experiments, we use L2 norm for our ball of radius $\rho$.

### H.4.2 Hyperparameter tuning for CIFAR-10 & CIFAR-100

We trained each model for 200 epochs and chose the model with highest validation accuracy on a held-out validation set. We used initial learning rate of 0.1 with learning rate decay of 0.2 at 60th, 120th, and 160th epoch with batch size of 128. We trained the models with weight decay 0.0005 and Nesterov momentum of 0.9. For SWA on CIFAR-100, we trained first 150 epoch with vanilla SGD and used weight averaging for the rest of the epochs.

### H.4.3 Hyperparameter tuning for Texas-100 & Purchase-100

We chose the best model as discussed before for CIFAR-10/100. We trained models with a learning rate of 0.1 with weight decay 0.0005 and Nesterov momentum of 0.9. We trained the models on Purchase-100 for a total of 100 epochs and on Texas-100 for a total on 75 epochs. During training, we employed a batch size of 512 for the Purchase-100 dataset and a batch size of 128 for the Texas-100 dataset.

### H.4.4 Hyperparameter tuning for EyePacs

We trained ResNet-18 with SGD, SAM and our proposed loss using EyePacs dataset for 100 epochs. Since, the dataset is highly imbalanced with about 25k data points out of 35k training data points belonging to one of the five classes, we used the balanced batch sampling strategy and a lower learning rate of 0.01 with learning rate decay of 0.2 at 60th epoch. As before, we also used weight decay 0.0005 and Nesterov momentum of 0.9. For our experiments, we utilized a batch size of 100, consisting of 12 samples from each of the 5 classes.

## I  Ablation Study: Comparison of different architectures

To validate consistency across different model architectures, we report direct threshold attack results in Table 6 using InceptionV4 [12] and ResNet-18 [13] for CIFAR-100 and CIFAR-10. We kept our $\rho$ the same across all model architectures with value 0.1. The results are consistent with our findings that SAM tends to have higher test accuracy while having higher membership attack accuracy at the same time. Overall best attack accuracy is higher for SAM for all the cases although we find mixed findings for multi-query attack accuracy specifically.

---

[12]https://github.com/weiaicunzai/pytorch-cifar100/blob/master/models/
[13]https://github.com/inspire-group/MIAdefenseSELENA/tree/main

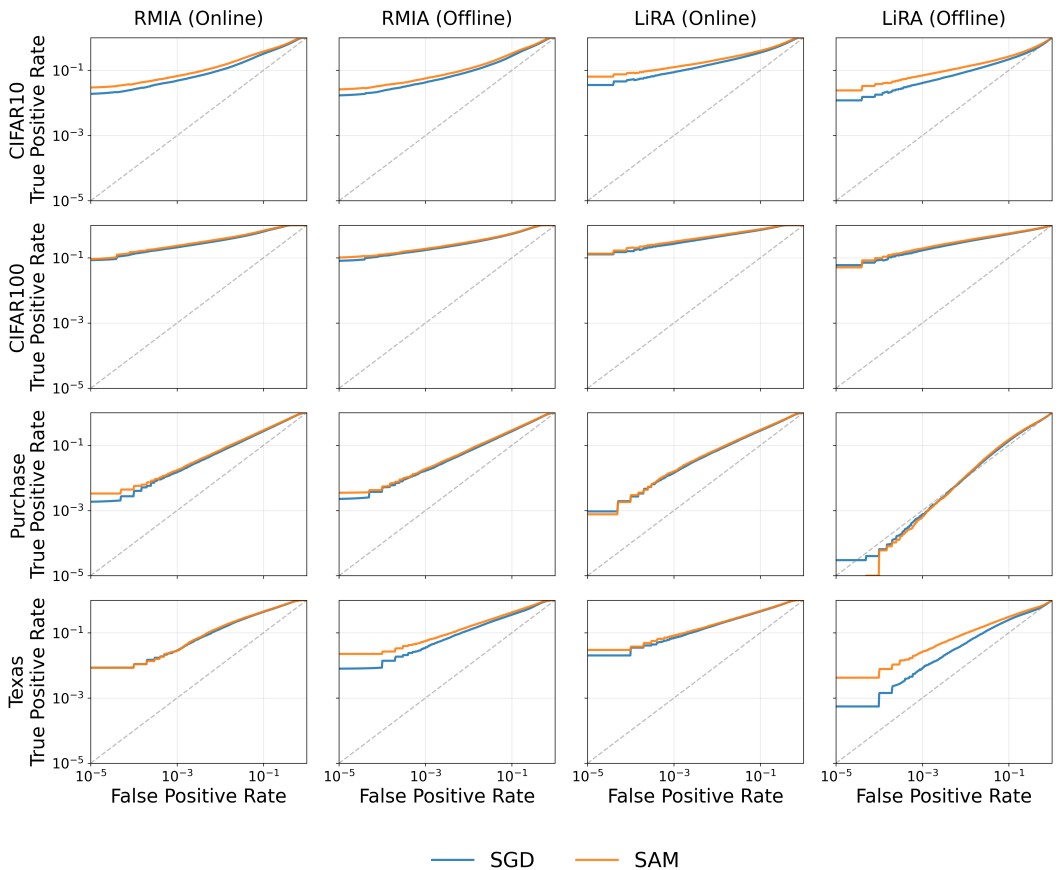

Figure 7: ROC curves comparing SAM (Orange) vs. SGD (Blue) across all datasets and attack modes on log-log scale. Rows represent datasets (CIFAR-10, CIFAR-100, Purchase100, Texas100). Columns represent the attack configuration. The ROC curve for SAM (orange) is above the ROC curve for SGD (blue) for nearly the entire range for most settings.

