# OpenReview forum: "Membership Privacy Risks of Sharpness Aware Minimization"
_ICLR.cc/2026/Conference — ICLR 2026 Poster_

### Official Review · Reviewer_g2BK · 2025-10-26

**Soundness:** 3
**Presentation:** 3
**Contribution:** 3
**Rating:** 8
**Confidence:** 3

**Summary:**

This paper studies the vulnerability of neural networks trained using algorithms that improve generalisation, such as the Sharpness-Aware Minimisation (SAM), to membership inference attacks (MIAs). It provides empirical evidence that suggests SAM-optimised networks are more vulnerable to MIAs when compared to SGD-optimised networks. The author(s) also provide theoretical justification for the high MIA vulnerability of SAM-optimised networks despite improved generalisation, since that goes against the conventional expectation where high MIA vulnerability is associated with low generalisation (and vice-versa).

**Strengths:**

- The paper presents evidence that suggests that optimising models using SAM encourages memorisation of atypical samples (mid-range memorisation scores), which contribute to their improved generalisation.
- The author(s)' proposed metric of influence entropy $\mathcal{I}_{ent}$ in Eq(6) helps verify the claim that SAM enhances generalisation by encouraging the network to memorization more atypical samples compared to SGD-based optimisation (as seen in Figure 4(a)).
- The empirical results show that the claims in the paper are not model-dependent but extend to different experimental settings.
- The theoretical justification for the observed phenomenon of improved generalisation leading to high memorisation when optimising a model with SAM is well described.

**Weaknesses:**

- Computing memorisation scores using Feldman and Zhang's [1] method is computationally expensive as it often requires (re)training hundreds of models. Why did the author(s) not use Ye et al.'s [2] more efficient version to compute these scores?
- Use of balanced accuracy as the attack metric and not TPR at fixed FPR [3], which informs more about an attack's ability to correctly identify membership signal at [preferably] low FPR (or low chances of predicting a member as a non-member).
- It does not use SOTA MIAs such as LiRA [3] / Quantile-MIA [4] / RMIA [5] to measure the sensitivity of SAM-optimised networks, which are known to provide better estimates of a model's MIA vulnerability compared to attacks that rely on predetermined thresholds, such as the Entropy- or Confidence-based attacks. There is also the case that the theoretical results for Theorem 2 depend on a single [data-dependent] threshold, whereas a factor contributing to the success of SOTA MIAs is that they incorporate sample-level thresholds.

[1] Feldman, V., and Zhang, C. “What Neural Networks Memorize and Why: Discovering the Long Tail via Influence Estimation.” NeurIPS 2020.

[2] Ye, J. et al. "Leave-One-Out-Distinguishability in Machine Learning." ICLR 2024.

[3] Carlini, N. et al. "Membership Inference Attacks From First Principles." SP 2022.

[4] Zarifzadeh, S. et al. “Low-Cost High-Power Membership Inference Attacks.” ICML 2024.

[5] Bertrán, M. et al. "Scalable Membership Inference Attacks via Quantile Regression." NeurIPS 2023.

**Questions:**

**Questions**: I would urge the author(s) to address the weaknesses detailed above. I am amenable to updating my initial assessment thereafter.

**Suggestions**: I suggest the following edits to improve the presentation of the paper:
- Minor Suggestion #1: Memorisation scores are measured w.r.t. samples, so this statement, "Motivated by this connection, we analyse the memorisation scores of SAM-trained models...", is somewhat misleading. It would be better to amend it to frame it w.r.t. individual samples.
- Minor Suggestion #2: Can you report the correlation coefficient between mem_SAM and mem_SGD for Figure 1(b) and 1(c)?
- Minor Suggestions #3: Lines 350-352 are written in a complicated and difficult-to-read manner. It would be best to rewrite it focusing on one relationship (for example, lower number bucket is associated with higher memorisation and vice versa).

---

> ### Author Response · Authors · 2025-11-21
>
> We thank the reviewer for taking the time to read our paper and providing a valuable feedback.
>
> **Memorization Score Computation**:
> Ye et al. estimate memorization/self-influence through Leave-One-Out Distinguishability (LOOD). They give an analytic LOOD estimator by modeling predictions with coupled Gaussian processes; for neural networks they instantiate this with the architecture’s NNGP kernel (infinite-width GP limit), avoiding expensive leave-one-out retraining. With the query set equal to the differing point, mean-distance LOOD in their GP/NNGP model coincides with Feldman–Zhang memorization, and empirically tracks retraining-based memorization for practical nets.
> However, because this proxy is determined by the architecture-induced kernel rather than the realized finite-width training trajectory, it is not aimed at capturing optimizer-specific effects such as SAM’s sharpness-aware perturbations. Our key phenomenon is optimization-level (SAM vs. SGD changes which samples are memorized/influential), so we use the direct retraining-based LOO definition. Moreover, our SGD baselines use Feldman et al.’s public CIFAR-100 memorization scores computed with the Feldman–Zhang retraining protocol, so applying the same definition to SAM is necessary for a controlled comparison. Mixing an NNGP proxy for SAM with retraining-based scores for SGD would confound the comparison. Overall, Ye et al.’s LOOD framework is highly valuable for fast architecture-level memorization estimation; our study instead targets optimizer-level mechanisms, so we adopt the retraining-based definition that directly reflects the learned SAM/SGD solutions.
>
>
> **Other MIAs and Metrics**:
> These are all valid issues. To address these concerns, we have implemented shadow model attacks including LiRA and RMIA and report the online attack results in Table 2 in the main text (or see our response to **2e1z**). The offline attacks are reported in Table 4 in the appendix. For the shadow model attacks, we report the AUC and TPR@0.1%FPR evaluation metrics in addition to balanced attack accuracy. Under these more advanced attacks, the results are consistent: SAM achieves superior generalization, but carries higher MIA risk. We find that the gap is more pronounced for Purchase100 and Texas100 datasets compared to previous attacks that only showed marginal difference.
>
> **New Results**:
>  We have updated our manuscript in a couple of different ways. The most profound update is that we added a new theory describing the mechanism of SAM driving MIA risk. We model SAM as a perfect interpolant under sharpness aware geometry and derive an inherent property of SAM: variance shrinkage. SAM lowers the variance of non member prediction compared to the euclidean interpolant (analogous to SGD). Using these results, we prove a higher MIA advantage for confidence threshold attack and likelihood ratio attack. As pointed out by the reviewer, we consider a single OUT distribution rather than a per-sample distribution for the attack. Inspired by the reviewer’s comment, we prove a lemma considering the per-sample case. We thank the reviewer for insightful comments.
> Second, we added two more sharpness aware optimizer variants to explore if the effects we discovered are only specific to SAM or applicable to other optimizers that look for flat minima. We tested on ASAM and GSAM, and observed a similar trend with SAM, suggesting that this privacy-utility tradeoff may be connected to flat minima in general.
>
>
> **Minor comments**:
> * We modified the sentence to “Motivated by this connection, we analyze the memorization scores of samples trained with SAM…”
> * We report pearson and spearman correlation coefficients between two memorization scores.
> *                 Correlation: 0.9603804639758303, P-value: 0.0
> *                 Spearman Rank: 0.9641683471696625, P-value: 0.0
>
> * We modified the sentence as recommended for clarity.
>
> We hope the reviewer is pleased by our updates and consider raising his/her score.

---

> > ### Comment · Reviewer_g2BK · 2025-11-24
> > **Follow-up Comment by Reviewer**
> >
> > I appreciate the authors' changes, and I do believe they are helpful towards improving the paper and strengthening its contributions. I will be maintaining my initial score.

---

### Official Review · Reviewer_2e1z · 2025-10-30

**Soundness:** 3
**Presentation:** 4
**Contribution:** 3
**Rating:** 6
**Confidence:** 4

**Summary:**

This paper reveals a critical trade-off in Sharpness-Aware Minimization (SAM): while it improves model generalization, it also increases privacy risk by making models more vulnerable to Membership Inference Attacks (MI). The authors demonstrate that this occurs because SAM's performance gains stem from its enhanced ability to memorize rare sub-patterns within the training data.

**Strengths:**

- The paper is well written and easy to follow.

- The finding of the relationship between the SAM and memorization is interesting and may shed light to privacy defense.

- The theoretical analysis is interesting and insightful.

**Weaknesses:**

+ (Major) Experimental Evidence. The central claim regarding SAM's heightened privacy risk is not yet fully convincing. The results in Table 1 show that SAM's privacy risk (measured by ASR) is not consistently or significantly higher than that of SGD (in Purchase-100 and Texas-100). Besides, Table 3 reports only the best attack accuracy, unlike Table 1 which shows results for all MIA methods.

+ (Major) Uncertain connection between ASR and memorization score. The link between the memorization score and the Membership Inference Attack (MIA) success rate is presented as a given. However, this relationship appears to be an assumption rather than an empirically demonstrated fact. This connection should be either validated or more cautiously framed as a hypothesis.

+ (Major) What can the observation bring to the future work on MIA or SAM? A more precise presentation of the study's potential implications for relevant fields would substantially strengthen the significance of this research.

+ (Minor) The solidness of the experiments would be strengthened by evaluating a wider range of modern MIA methods to ensure the findings are not specific to the selected attacks.

+ (Minor) The experiments are primarily conducted on small datasets. It is unclear if the observed privacy trade-offs persist on larger, more complex benchmarks such as ImageNet.

## Score Justification

The paper is well-organized and introduces valuable theoretical analysis. However, the experimental evidence currently feels incomplete, which undermines the strength of its conclusions. Besides, the most significant shortcoming is the lack of a clear discussion on the broader impact and implications of these findings for the community. While briefly mentioned in the conclusion, a detailed discussion on how this new understanding of SAM's properties (especially concerning long-tailed data) can influence future research and practice is crucial.

**Questions:**

See the weaknesses.

---

> ### Author Response · Authors · 2025-11-21
>
> We thank the reviewer for taking the time to read our paper and providing a valuable feedback.
>
> **Experimental Evidence**:
> To provide a more rigorous empirical verification, we employed shadow model attacks including LiRA and RMIA. The results for online attack are in Table 2 of the updated manuscript, but we also include the results here. For the new experiments, we report AUC and TPR@0.1%FPR metrics in addition to ASR. The results show that even under shadow model attacks, SAM incurs higher membership privacy risk despite better generalization. Because the data split and experimental setup is different from the previous set of attacks, the shadow model results alleviate the reviewer’s concern about less obvious gaps for Purchase100 and Texas100 datasets. Under shadow model attacks, the gap is much more pronounced. Additionally, we report results for two other variants of sharpness aware optimizers (ASAM and GSAM) to explore if this phenomenon is specifically applicable to SAM algorithm or also applies to optimizers that seek a flatter minima. The results are reported in Table 3 in the appendix. We hope the reviewer is pleased by these refinements.
>
> | Dataset | Attack | SGD Test Acc | SGD AUC | SGD Att | SGD TPR | SAM Test Acc | SAM AUC | SAM ASR | SAM TPR |
> | :--- | :--- | :---: | :---: | :---: | :---: | :---: | :---: | :---: | :---: |
> | **CIFAR-100** | RMIA | 67.7% | 90.4% | 81.2% | 18.0% | **69.2%** | **91.5%** | **82.6%** | **21.8%** |
> | | LiRA | 67.7% | 92.0% | 82.1% | 25.4% | **69.2%** | **93.2%** | **83.6%** | **28.3%** |
> | **CIFAR-10** | RMIA | 92.3% | 71.4% | 63.7% | 7.4% | **93.1%** | **74.7%** | **65.9%** | **10.4%** |
> | | LiRA | 92.3% | 71.6% | 63.5% | 7.0% | **93.1%** | **75.5%** | **66.2%** | **10.6%** |
> | **Purchase100** | RMIA | 75.3% | 66.9% | 61.9% | 1.2% | **78.2%** | **70.2%** | **64.2%** | **1.6%** |
> | | LiRA | 75.3% | 66.2% | 61.1% | 0.4% | **78.2%** | **69.7%** | **63.3%** | **1.3%** |
> | **Texas100** | RMIA | 46.8% | 67.5% | 64.5% | **0.6%** | **50.1%** | **73.7%** | **69.7%** | 0.0% |
> | | LiRA | 46.8% | 80.1% | 71.0% | 5.7% | **50.1%** | **81.2%** | **71.8%** | **7.5%** |
>
> **Connection between memorization and MIA**:
> We agree with the reviewer that in the previous version, this connection may have not been specifically explained. In the updated manuscript, we have updated the content to discuss the concrete mechanism or inherent property of SAM that is causing higher MIA risk. We claim that SAM has an inherent property of reducing output variance. This effect prevents the model from becoming too heavily dependent on majority features and learning more diverse features (minority subclass alignment). We added a new theory modeling SAM as a perfect interpolant under sharpness aware geometry to prove this property. Then, we show how lower variance provably increases MIA advantage for confidence threshold attack and likelihood ratio attack.
>
> **Future Work**:
> We believe one implication would be studying loss geometry and its relationship to MIA attacks. We reported a custom loss optimizer Sharp. The results showed that opposite to SAM, Sharp exhibited decreased generalization but at the same time lower privacy risk. One crucial perspective of our work is the role of variance shrinkage. We believe further scrutiny could help develop effective MIA defense mechanisms. Moreover, previous works have reported that SAM tends to learn simple features [1]. Implications from memorization and influence scores in our work, however, seem to point in the opposite direction: SAM is better at learning diverse features. These unconventional findings would benefit the community.
>
> **Datasets**:
> The datasets we employed are part of standard benchmarks in MIA research. We thank the reviewer for helpful comments. We hope the reviewer is satisfied with our updates and consider raising his/her score.
>
> [1] Andriushchenko et al. Sharpness-Aware Minimization Leads to Low-Rank Features (NIPS '23)

---

> > ### Comment · Reviewer_2e1z · 2025-11-24
> >
> > Thanks for your reply. I think the **Future Work** is valuable for the community. I also agree that more advanced MIA method should be evaluated as the other reviewers comment. Therefore, I keep my score as 6.

---

> > > ### Author Response · Authors · 2025-11-24
> > >
> > > Thank you for your follow-up and for recognizing the value of our future work.
> > >
> > > We would like to kindly clarify that, in response to the initial reviews, we have incorporated results for **more advanced** LiRA and RMIA as asked by the other reviewer in the main section of our paper. the results we added confirm our general finding: SAM incurs higher membership privacy risk despite achieving better generalization. our revised manuscript now includes six different types of MIA evaluations (confidence-based, entropy-based, modified entropy, NN-based, LiRA, and RMIA), covering both threshold-based and shadow-model attacks.
> > >
> > > We would also like to emphasize the critical importance of our paper: while many works promote the idea that “flatness = good,” our study provides the first cautionary evidence that this assumption is not universally true, improved generalization through flatter minima can come at the cost of increased privacy risk. Given the growing adoption of SAM in practice, we believe these findings are critical for the community and warrant attention.
> > >
> > > Since we have addressed all comments and concerns raised during the review process, we would greatly appreciate if you agree and raise your score accordingly.

---

> > > > ### Comment · Reviewer_2e1z · 2025-11-26
> > > >
> > > > Thanks for your clarification. I apologize for any ambiguity in my earlier comments—what I intended to convey was that more advanced MIA methods ideally should have been evaluated in the first revised manuscript.
> > > >
> > > > Regarding the score, I'd like to emphasize that it reflects my overall assessment of the paper’s quality. Addressing all of my concerns does not automatically lead to a score increase, as the rating takes into account the work in its entirety.
> > > >
> > > > Overall, I continue to find this an interesting paper that offers a fresh perspective on SAM and privacy attacks. However, I still have reservations regarding the experimental results, which, in my view, are not yet strong enough to fully support the paper’s central claim (the ASR gap between SAM and SGD is not very significant, in most datasets less than 3%).

---

> > > > > ### Author Response · Authors · 2025-11-28
> > > > >
> > > > > Thank you for your constructive feedback and continued engagement. We totally agree that the rating should reflect the overall quality of the results in the paper. Below, we will make short, concrete case for why we are a bit surprised at the lukewarm response to the paper:
> > > > >
> > > > > **Generalization–Privacy Trade-off & Optimizer Geometry**: Our work is the first to reveal a geometry-mediated privacy–utility trade-off: optimizers seeking flatter minima (SAM, GSAM, ASAM, SWA) for better generalization consistently increase membership leakage. Given the growing popularity of SAM (as well as its variants discussing how SAM learns better features [1] and favours simplicity bias [2]), our paper acts as a cautionary tale about the “flatness = good” narrative and is highly relevant. We provided both theoretical evidence of why this happens (Theorems 1,2,3 -- discussing variance-shrinkage and subgroup generalization) as well as empirical validation across six MIA variants (including state-of-the-art LIRA/RMIA) with multiple metrics (Attack Accuracy, AUC and TPR at 0.1% FPR) on standard benchmarks and architectural ablations.
> > > > >
> > > > > **Privacy gaps are meaningful**: While the ASR gap may appear modest (3\%), it mirrors SAM’s generalization gains, which prior work considers significant [3]. Moreover, SAM consistently yields higher AUC and more member exposure at low FPR than SGD across all datasets. These differences confirm that the privacy risk increase is meaningful, not just a statistical blip. We are not claiming a catastrophic failure of SAM but rather a clear systemic shift in risk wherever SAM gains utility, which is worth sharing with the community.
> > > > >
> > > > > **Beyond dataset level analysis**: We also dissect SAM’s behavior at the datapoint level, asking and answering for which datapoints SAM gains the most. This is of independent interest as these theoretical understandings of SAM will act as a guiding light to designing new algorithms that seek a better equilibrium on the spectrum of privacy vs generalization tradeoff. Our results demonstrate that manipulating the loss landscape geometry can significantly influence this tradeoff, and we are the first to do so.
> > > > >
> > > > > Once again, thank you for pushing us to improve the paper. We have worked diligently to make the contribution clear and solid. With the new changes in the manuscript, we believe our findings are not just an isolated curiosity but rather a noteworthy consideration for the community. We hope the reviewer agrees.
> > > > >
> > > > > [1]. Springer, Jacob Mitchell, Vaishnavh Nagarajan, and Aditi Raghunathan. "Sharpness-aware minimization enhances feature quality via balanced learning." ICLR 2024.
> > > > >
> > > > > [2]. Chang, Wei-Kai, and Rajiv Khanna. "A Unified Stability Analysis of SAM vs SGD: Role of Data Coherence and Emergence of Simplicity Bias." NeurIPS 2025.
> > > > >
> > > > > [3]. Foret, Pierre, et al. "Sharpness-aware Minimization for Efficiently Improving Generalization." ICLR 2021.

---

### Official Review · Reviewer_U9Jc · 2025-10-31

**Soundness:** 3
**Presentation:** 3
**Contribution:** 3
**Rating:** 6
**Confidence:** 3

**Summary:**

This paper presents a counterintuitive yet rigorously studied phenomenon: Sharpness-Aware Minimization (SAM), an optimization method known for improving generalization, unexpectedly increases vulnerability to membership inference attacks (MIAs). Through extensive experiments on multiple datasets and theoretical analysis, the authors demonstrate that SAM’s generalization gains stem from structured memorization of atypical subclass patterns(e.g., rare features in long-tailed distributions), which simultaneously enhances test performance and privacy risks. The work challenges the conventional belief that better generalization implies lower privacy risk, offering novel insights into the trade-offs between optimization, generalization, and privacy.

**Strengths:**

1. The paper offers novel insights into the trade-offs between generalization and privacy, which challenges the conventional belief that better generalization implies lower privacy risk.
2. The authors combine empirical evidence with theoretical guarantees to support their claims. The consistent results across datasets and models strengthen their conclusion.
3. The paper is well-written and easy to follow.

**Weaknesses:**

1. While the paper highlights SAM’s privacy vulnerability, it would be good to propose or evaluate defensive strategies to mitigate this risk to improve practical applicability.
2. The theoretical analysis relies on a simplified linear model and strong assumptions (e.g., perfect interpolation). It lacks both theoretical and empirical validations on more advanced non-linear architectures like Transformer-based or diffusion models.

**Questions:**

See the weaknesses.

---

> ### Author Response · Authors · 2025-11-21
>
> We thank the reviewer for taking the time to read our paper and providing a valuable feedback.
>
> **Defensive strategies and mitigation**: We agree with the reviewer that developing defenses is a crucial next step for practical applicability. In our manuscript, we have introduced a custom designed, "Sharp" optimizer (now in Table 3 of the Appendix). This optimizer, which explicitly seeks sharper minima, acts as a counter-study to SAM. Our results show that while the Sharp optimizer significantly reduces MIA vulnerability (acting as a successful defense), it does so at the cost of generalization. This reinforces our core finding of a fundamental utility-privacy trade-off mediated by the geometry of the loss landscape. A completely novel defense that achieves better tradeoff than above and its evaluation will require significant empirical evidence, and hence is a separate useful future work beyond the scope of our current work. We hope the reviewer agrees that our novel contributions on unearthing the link between flatter minima and privacy leakage which provide the necessary geometric understanding (variance shrinkage) to guide defensive mechanisms deserves attention to encourage such future works.
>
> **Theory**: Please see response to reviewer **4axt** for deeper discussion of theory. We have also added  a more general theorem. We model SAM as a perfect interpolant under sharpness aware geometry and prove an inherent property of SAM: variance shrinkage. By lowering the variance of model output, the prediction on non members and members become more separable, leading to higher membership privacy risk. Our theory now connects SAM to variance reduction, and to higher attack advantage for confidence and likelihood ratio attack. This result does not need the anti-alignment assumption even though we do believe it is valid (please see response to reviewer 4axt for further elaboration on this).  Perfect interpolation is a very natural assumption in studying neural networks as they are known to easily reach near zero training error.
> We do acknowledge that the linear interpolating regime is a simplified model. However, this setting is the standard theoretical framework for analyzing the behavior of overparameterized neural networks (e.g., the Neural Tangent Kernel regime). By using this framework, we are able to mathematically isolate the specific effect of geometry (sharpness) on prediction variance without the confounding factors of non-convex optimization dynamics. Our new theorem rigorously proves that even in this simplified setting, minimizing sharpness inherently reduces the variance of non-member predictions, providing a clear mechanistic explanation for the empirical vulnerability observed in deep networks.
> Regarding Transformers and diffusion models:
>
> * Transformers: We believe the mechanism we identified—variance shrinkage in flat minima—is architecture-agnostic. The correlation between sharpness aware optimization and generalization is documented in Transformers as well [1]. We hope our work encourages further evaluations on transformers too.
> * Diffusion Models: While extremely interesting, MIA on generative diffusion models requires fundamentally different attack formulations (e.g., analyzing generated image fidelity rather than classification confidence), which merits a separate dedicated study
>
> We hope the reviewer is satisfied with our updates and consider raising his/her score.
>
> [1] IIbert et al. SAMformer: Unlocking the Potential of Transformers in Time Series Forecasting with Sharpness-Aware Minimization and Channel-Wise Attention (ICML '24)

---

### Official Review · Reviewer_4axt · 2025-11-01

**Soundness:** 2
**Presentation:** 3
**Contribution:** 2
**Rating:** 2
**Confidence:** 4

**Summary:**

This paper challenges the conventional belief that improved model generalization implies better membership privacy by investigating the SAM algorithm. The authors surprisingly find that SAM-trained models, despite achieving better generalization than standard SGD, may be more vulnerable to MIAs. The authors hypothesize this occurs because SAM selectively learn atypical but generalizable sub-patterns more effectively than SGD. This enhanced memorization of rare features improves performance on atypical test samples, but consequently increases the privacy risk by making training samples more distinguishable. The paper empirically validates this mechanism by analyzing memorization and influence scores and provides a theoretical framework showing how a model that better captures minority features can simultaneously achieve high generalization and high MIA vulnerability.

**Strengths:**

1. The paper first states that SAM increases vulnerability to MIAs.
2. The paper is exceptionally clear, presenting its counter-intuitive finding and complex hypothesis in a logical, well-structured, and easy-to-follow manner.

**Weaknesses:**

1. The authors claim to challenge the conventional assumption that improved generalization implies stronger privacy. However, this insight has already been discussed in previous works [1,2,3,4].
2. The paper's claims are based solely on the original SAM algorithm. It is unclear whether these findings on increased privacy risk generalize to other sharpness-aware optimizers (e.g., ASAM, GSAM). An investigation into these variants is recommended.
3. As a privacy metric, accuracy has been criticized in many works [4,5]. It is recommended to provide TPR at a low FPR and evaluate the privacy risk using recent MIA methods, such as LiRa and RMIA [4,6].
4. The "anti-alignment" assumption means a model that gets better at classifying the majority subclass inherently gets worse at classifying the minority subclass, forcing a direct trade-off. This is a very strong and specific setup.



[1] "Understanding membership inferences on well-generalized learning models" (arXiv, 2018)

[2] "When does data augmentation help with membership inference attacks" (ICML, 2021)

[3] "Bounding Information Leakage in Machine Learning" (Neurocomputing, 2023)

[4] "Membership Inference Attacks From First Principles" (SP 2022)

[5] "Evaluations of Machine Learning Privacy Defenses are Misleading" (CCS, 2024)

[6] "Low-Cost High-Power Membership Inference Attacks" (ICML, 2024)

**Questions:**

I do not understand why Assumption 4 is reasonable. Could you explain it?

---

> ### Author Response · Authors · 2025-11-21
>
> We thank the reviewer for taking the time to read our paper and providing a valuable feedback.
>
> **Connection between generalization and MIA risk**: We thank the reviewer for highlighting these foundational works. We have updated our introduction (Line 43) to explicitly discuss [1-4] suggested by the reviewer. While we acknowledge the known tension between privacy and overfitting, our work contributes a novel perspective by isolating the specific mechanism of Sharpness-Aware Minimization (SAM). Specifically, we demonstrate that even when generalization improves, the specific geometry of flat minima can inadvertently increase privacy risks, which none of the previous works address.
>
> **Other sharpness-aware algorithms**: We appreciate the suggestion to broaden the scope of our analysis. We have performed additional experiments on ASAM (Adaptive SAM) and GSAM (Surrogate Gap SAM). These results are now included in Table 3 (Appendix). Consistent with our findings on standard SAM, these variants also exhibit improved generalization alongside increased vulnerability to MIA. This confirms that privacy leakage is a fundamental characteristic of sharpness-aware optimization strategies, not just an artifact of the original SAM algorithm.
>
> **Extending experiments**: Following the reviewer’s recommendation, we have expanded our evaluation to include state-of-the-art shadow model attacks, specifically LiRA and RMIA. We now report online attack AUC and TPR @ 0.1% FPR in Table 2 of the main text (or see response to **2e1z**) and offlline attack results in Table 4 in the appendix. The results demonstrate that SAM’s increased vulnerability persists even under these rigorous low-false-positive metrics, confirming a robust utility-privacy tradeoff. Also, we note that under shadow model attacks the difference in privacy gap is more pronounced for Purchase100 and Texas100 datasets.
>
> **Anti-alignment assumption**:
> In many real scenarios, a minority subgroup’s examples might contain a feature that “interferes” with or opposes the main feature used for the majority of cases. A wide range of prior empirical and theoretical works support the plausibility and usefulness of the anti-alignment assumption: (a) formal connections have been drawn between spurious correlation bias and subgroup performance [4]. Many robustness benchmarks (like Waterbirds and CelebA) explicitly construct such feature misalignment: e.g., most training birds appear with a particular background (water vs. land), while rare instances appear in the opposite background. A standard ERM-trained model picks up the background as the deciding feature (since it holds for the majority), and consequently flips the labels for the minority subgroup, (b) Learning theory on long-tailed distributions shows that rare subpopulations must be memorized for optimal generalization because they do not align with the majority populations [1,2]. (c) Max-margin generalization in presence of noisy anti-aligned rare data points essentially requires over-parameterization to carve out design boundaries around these rare data points which can not be explained by majority margins and simpler models just catering to the majority class will misclassify them [3]. All these insights reinforce the anti-alignment scenario: when a minority group’s key feature is anti-correlated with the majority’s feature, a standard ERM learner will do well on the majority and systematically err on the minority – unless it effectively memorizes or learns a separate rule for that subgroup.
>
> Although we believe that our setup is not narrow, our results do hold in more general settings too. We have replaced the previous MIA theorem with a more general theoretical framework that does not depend on anti-alignment or assumption 4.
> * We now analyze the Variance Shrinkage property of SAM in the interpolating regime.
> * We model SAM as a perfect interpolant under sharpness-aware geometry and compare it to the Euclidean min-norm solution (SGD).
> * New Theorem: We prove that SAM inherently lowers the prediction variance on non-member data compared to SGD and that this amplifies the advantage of both confidence and likelihood ratio attacks.
>
> We have moved the analysis regarding subclass data to the Appendix, as it provides a useful intuitive link between subclasses and generalization in light of the presented use-cases of anti-alignment above.
> We hope the reviewer is pleased with our updates and considers raising his/her score.
>
> [1] Feldman, Does Learning Require Memorization? A Short Tale about a Long Tail (STOC '20)
>
> [2] Brown et al. When is memorization of irrelevant training data necessary for high-accuracy learning? (STOC '21)
>
> [3] Chatterjee et al. Finite-sample Analysis of Interpolating Linear Classifiers in the Overparameterized Regime (JMLR '21)
>
> [4] Mehta. When Are Learning Biases Equivalent? A Unifying Framework for Fairness, Robustness, and Distribution Shift (eurIPS '25 workshop)

---

> > ### Comment · Reviewer_4axt · 2025-11-27
> >
> > Thank you for your response; it has addressed some of my points. However, I still have the following concerns:
> >
> > 1. RMIA is reported to achieve SOTA attack performance, while in your experimental results in Table 2, it seems that LiRA is better. Can you explain this discrepancy? Furthermore, I recommend including the ROC curve on a logarithmic scale for a more comprehensive evaluation.
> > 2. The claim that "higher membership privacy risk is associated with poor generalization" is **an older perspective that has already been questioned**. Therefore, there is no need to claim you are "challenging the conventional assumption that improved generalization implies stronger privacy.", as this is not a surprising finding. In addition to the literature I mentioned previously, experimental results from RelaxLoss [1] and CCL [2] also show that test accuracy and privacy risk often increase simultaneously.
> > 3. I note that from your assumption, you derive $S_{G_\eta}(X_{\mathrm{in}}) \sim \mathcal{N}(0, v_{\mathrm{in}})$ and $S_{G_\eta}(X_{\mathrm{out}}) \sim \mathcal{N}(0, v_{\mathrm{out}}(G_\eta))$, which implies the scores for members and non-members are equivalent in expectation. I suspect this intermediate result is unrealistic for the MIA scores of deep learning models.
> >
> > [1]  "RelaxLoss: Defending Membership Inference Attacks without Losing Utility" (ICLR2022)
> >
> > [2]  "Mitigating Privacy Risk in Membership Inference by Convex-Concave Loss" (ICML2024)

---

> ### Author Response · Authors · 2025-12-02
>
> Thank you for the engagement in the discussion.
>
> **Performance of RMIA vs LiRA**: Thank you for pointing this out. This discrepancy was partly because of number shadow models. We have increased the number of shadow models to 256, which now matches the settings in [1,2]. The updated results are reported in Tables 2 and 4, showing improved TPR@0.1%FPR for RMIA. **Our main results about SAM consistently exhibiting higher membership leakage still hold**.
>
> In our revised experiments we observe the following:
> * In the offline setting (where RMIA was designed to be particularly strong), RMIA indeed achieves substantially higher AUC than LiRA, consistent with [1].
> * In the online setting, the gap between RMIA and LiRA is small, but we observe that LiRA is slightly stronger than RMIA. We found that the online RMIA performance is quite sensitive to the choice of its hyperparameters (e.g., $\gamma$ and $\alpha$), whereas LiRA has fewer attack-specific knobs to tune. Since our focus is not on proposing a new SOTA attack but on comparing the relative privacy of SGD vs. SAM under strong attacks, we emphasize that our conclusions are robust across both LiRA and RMIA rather than claiming RMIA is always the strongest attack in our setup. We have revised the discussion around Table 2 in the appendix accordingly to explicitly acknowledge this. Our main empirical message is that **for almost all datasets and all attacks considered, SAM consistently exhibits higher membership leakage than SGD**, even when the absolute strength of RMIA vs LiRA varies by setting.
> We have also followed your suggestion and added ROC curves on a log-log scale (see new Figure 7 in the appendix) to give a more complete view of the attack behavior. Figure 7 shows that SAM's ROC curve dominates that of SGD over nearly the entire range across most setups. This corroborates the strong result from our theoretical proofs showing that **SAM can incur higher MIA advantage over any fixed threshold**.
>
> **Contribution statement**: We appreciate the references to RelaxLoss and CCL, which are membership privacy defenses. We do acknowledge that the generalization-privacy trade-off is established in the context of privacy defenses. However, our contribution lies in identifying this phenomenon within standard optimization rather than defense mechanisms. While defenses intentionally manipulate this trade-off, SAM is designed solely to improve generalization. We appreciate the reviewer’s suggestion regarding our phrasing,, we have removed the term "conventional wisdom." Also, we have updated the Abstract and the Introduction to tone down our contribution about generalization-privacy tradeoff. Rather than claiming to “challenge the conventional wisdom”, we will clarify that our paper “**uncovers the very geometric mechanism of SAM that improves generalization and simultaneously exacerbates membership leakage**.” This revision should clearly convey that our focus is on the *specific cause* of the tradeoff in SAM-trained models, without overstating the novelty of the high-level observation. This is especially interesting because the cause of generalization performance of SAM is still an unresolved question with recent works attributing it to either SAM’s implicit bias towards diversity [3] or simplicity [4,5] of features. Lastly, we expanded our contribution as a cautionary tale against *flatter minima=good* notion prevalent in many works, but from a privacy standpoint.
>
> **Assumption of $S_G$**: We thank the reviewer for highlighting this point, as it allows us to clarify a confusion in our notation. In our theoretical framework, the variable $S_G(x)$ was defined as the raw, signed model output (i.e., $f_G(x)$), rather than the final membership score. For likelihood ratio attack setup, we state that the attacker uses $S_G(x) = f_G(x)$, which is simply the output of the model. Hence, the mean zero assumption means that data samples are distributed symmetrically centered at zero, which is a natural setting for binary classification, linear regression and SVMs. Due to the two-sidedness of the LR test, eventually what is evaluated is the absolute value of $S_G$, identical to the confidence attack. We never assume $|S_G|$ is mean zero for member and non-members, as it would be unrealistic as the reviewer pointed out. To avoid this confusion, we have removed the additional notation of $S_G$ and modified our manuscript to use $f_G$ all around.
>
> [1]Carlini, N. et al. "Membership Inference Attacks From First Principles." SP 2022.
>
> [2]Zarifzadeh, S. et al. “Low-Cost High-Power Membership Inference Attacks.” ICML 2024.
>
> [3]Springer et al. "Sharpness-aware minimization enhances feature quality via balanced learning." ICLR 2024.
>
> [4]Chang and Khanna. "A Unified Stability Analysis of SAM vs SGD: Role of Data Coherence and Emergence of Simplicity Bias." NeurIPS 2025.
>
> [5]Andriushchenko et al. “Sharpness-Aware Minimization Leads to Low-Rank Features.” Neurips 2023.

---

### Author Response · Authors · 2025-12-03

Dear AC and reviewers,

Thank you for all the constructive feedback and discussions, allowing us to improve our manuscript significantly. We summarize the revisions we made to the paper that addresses all the major suggestions proposed by the reviewers.

* **Abstract and Introduction**: We have toned down our contribution of reporting generalization vs privacy tradeoff and expanded our contribution as being a cautionary tale against the *flatter minima=good* notion from a privacy standpoint. We also introduce the existing tension between whether SAM’s implicit bias is geared towards diversity or simplicity, for which our findings align with the former. We have also added all the references suggested by the reviewers.

* **Section 3 - MIA experiments**: We have added results for SOTA membership inference attacks, specifically RMIA and LiRA, and report AUC, average attack accuracy, and TPR@0.1%FPR in Table 2 and 4 as suggested by the reviewers. We used 256 shadow models to match the settings of [1,2] for reliability. Discussions around the experimental setup and the results have been updated for the new results. New results are consistent with the previous findings, demonstrating SAM’s MIA vulnerability across almost all settings.

* **Section 4 - memorization and influence experiments**: We have added a discussion connecting our experiments on memorization and influence scores with SAM’s capability to learn more diverse features.

* **Section 5 - theoretical analysis**: We have introduced a new theorem showing SAM’s inherent property of variance reduction for non-member samples, and how this property leads to higher MIA advantage for confidence and likelihood ratio attacks. Our theoretical analysis not only shows higher advantage at optimal threshold, but at **any** fixed threshold indicating ROC curve dominance. We have modified and moved the previous theorem to the appendix addressing reviewers’ concerns regarding strong assumptions.

* **Conclusion and future works**: We have added more discussion of future works to address the reviewer’s suggestion of including future implications.

* **Appendix**: More related works and details about the new experiments have been included. Experimental results for other sharpness-based optimizers (ASAM and GSAM) and ensuing discussions have been added to Table 3 and Appendix F. Lastly, Figure 7 is now introduced, showing the ROC curves for SGD and SAM on a log-log scale, corroborating our theoretical result.

[1]Carlini, N. et al. "Membership Inference Attacks From First Principles." SP 2022.

[2]Zarifzadeh, S. et al. “Low-Cost High-Power Membership Inference Attacks.” ICML 2024.

---

### Author Response · Authors · 2025-12-03

Dear AC,

Thank you for taking the time to go through our work and the discussions. We have addressed all the concerns or issues raised by the reviewers. Below, we summarize the major points in the reviews and the discussions.

**Reviewer g2BK - initial score 8, acknowledged our response, retained score**:

* Complimented that our theoretical justification *is well described* and our experimental results corroborate our claims.

*  Asked for other MIAs and metrics. We have updated our experiments to include RMIA and LiRA using a large number of shadow models (256 matching the original papers) and now report AUC and TPR@0.1%FPR as additional metrics. Furthermore, we report averages of 10 different attack splits, adding more reliability to our results. The new results fully corroborate our claim.

* To our response, the reviewer acknowledged that the updates *are helpful towards improving the paper and strengthening its contributions*.


**Reviewer 4axt - initial score 2, positive trend**:

* Complimented that our paper is *exceptionally clear, presenting its counter-intuitive finding and complex hypothesis in a logical, well-structured, and easy-to-follow manner*.

* Concerned about our claim of challenging the conventional assumption that improved generalization implies stronger privacy is too strong and an overclaim We added references and discussion about this point in the paper and toned down our contribution in this regard, focusing more on the contribution that our paper uncovers **the very geometric mechanism of SAM that improves generalization and simultaneously exacerbates privacy leakage**. We have also expanded our contribution to serving as a cautionary tale against  *flatter minima = good* notion from a privacy standpoint.

* Asked for other MIAs and metrics. We have updated our experiments to include RMIA and LiRA using a large number of shadow models (256 matching the original papers) and now report AUC and TPR@0.1%FPR as additional metrics. New results fully corroborate our claim.

* Stated that assumption was too strong about our theory. To alleviate this issue, we replaced the relevant part of the previous theory with a new theory that fully connects sharpness-aware geometry to variance reduction, leading to higher MIA advantage. The new theory does not depend on the assumption mentioned by the reviewer. The reviewer further asked about an assumption for the new theory, but we believe it came from confusion of notation. Otherwise, the assumption is standard for binary classification, linear regression, and SVMs.


**Reviewer U9Jc - initial score 6, no response**:

* Complimented that our paper is *well-written and easy to follow* and that *consistent results across datasets and models strengthen our conclusion*.

* Asked about extension to a new MIA defense method. We introduced optimization that searches for sharper minima in our original manuscript, which indirectly addresses this to a certain extent. A complete evaluation would require much more experiments and would be another work on its own.

* Asked about extension to larger models such as transformers and theoretical analysis of non-linear models. As sharpness-aware optimization has been shown effective for transformers as well, we believe our results would apply as well. Furthermore, we justify that our theoretical setting is one of standard theoretical frameworks for analyzing the behavior of overparameterized neural networks.

**Reviewer 2e1z - initial score 6, acknowledged our response, retained score**:

* Complimented that our paper is  *well-written and easy to follow* and that our theoretical analysis is *interesting and insightful*.

* Asked for more empirical validation and other metrics. We have updated our experiments to include RMIA and LiRA as mentioned before.

* Asked for a more detailed future impact of our work on MIA and SAM. Regarding MIA, our introduction of Sharp optimization indicates the potential of sharper minima as a new defense mechanism. For SAM, our results challenge the notion that flatter minima=good from a privacy standpoint and that SAM has an implicit bias towards simplicity. We have added these future directions in the conclusion.

* The reviewer acknowledged that we have fully addressed the reviewer’s concerns. The reviewer’s remaining doubt was that the difference in MIA metrics between SAM and SGD was not large enough in his/her standards (near 3% difference). We justified that the generalization performance difference between SAM and SGD is about the same amount, and the consistency of higher MIA vulnerability over extensive number of settings validates our claim. Moreover, our strong theoretical result shows higher MIA advantage of SAM for not just optimal threshold but **any** fixed threshold. This is corroborated with Figure 7 in the appendix, illustrating how SAM’s ROC curve is above that of SGD for nearly the **entire** range, reaffirming the significance of our findings.

---

### Meta-Review · Area_Chair_qGdz · 2026-01-06

**Summary:**

- Reviewers questioned the strength of the empirical evidence, noting that the observed privacy gap between SAM and SGD was often modest and not consistently significant across all datasets.
- They also critiqued the reliance on simplified theoretical assumptions (e.g., perfect interpolation, anti-alignment) and suggested the paper’s claim of challenging the generalization-privacy trade-off was overstated, as prior work had already explored similar tensions.
- Other concerns include the need for evaluation with state-of-the-art membership inference attacks, more comprehensive metrics, validation on larger models, and a clearer discussion of the broader implications for both SAM and privacy defenses.

The authors' rebuttal has addressed most of the key concerns, so I recommend Accept.

**Reviewer Concerns:**

The authors’ rebuttal effectively addressed many of these concerns by expanding many experiments. They revised their analysis to focus on SAM’s variance reduction property without relying on the strong anti-alignment assumption. They also tempered their claims regarding challenging conventional wisdom and reframed the contribution as uncovering SAM’s specific geometric mechanism that improves generalization while exacerbating privacy leakage. However, some concerns may remain outstanding, particularly regarding the practical significance of the observed privacy gaps.

**Reviewer Scores:**

Reviewer g2BK (initial score: 8): Likely would have maintained the score.

Reviewer 4axt (initial score: 2): Would likely have increased the score to a 4 or 6, given the positive engagement and substantial revisions on the theory and assumption.

Reviewer U9Jc (initial score: 6): Might have maintained the score, as the authors addressed the need for defensive insights and provided a more general theoretical analysis, though extension to non-linear models remains limited.

Reviewer 2e1z (initial score: 6): Would probably have kept the score at 6, as they continued to express reservations about the experimental significance of the privacy gap despite the added experiments and metrics.

---

### Decision · Program_Chairs · 2026-01-26

Accept (Poster)